# AAV-delivered muscone-induced transgene system for treating chronic diseases in mice via inhalation

Xin Wu [1,2,7], Yuanhuan Yu [1,7], Meiyan Wang [1,3,7], Di Dai[1], Jianli Yin[1,3], Wenjing Liu [1], Deqiang Kong [1], Shasha Tang[4], Meiyao Meng[1], Tian Gao[1], Yuanjin Zhang[1], Yang Zhou [1,5], Ningzi Guan[1], Shangang Zhao [6] & Haifeng Ye [1,3,5] ✉

Gene therapies provide treatment options for many diseases, but the safe and long-term control of therapeutic transgene expression remains a primary issue for clinical applications. Here, we develop a muscone-induced transgene system packaged into adeno-associated virus (AAV) vectors (AAV$_{MUSE}$) based on a G protein-coupled murine olfactory receptor (MOR215-1) and a synthetic cAMP-responsive promoter (P$_{CRE}$). Upon exposure to the trigger, muscone binds to MOR215-1 and activates the cAMP signaling pathway to initiate transgene expression. AAV$_{MUSE}$ enables remote, muscone dose- and exposure-time-dependent control of luciferase expression in the livers or lungs of mice for at least 20 weeks. Moreover, we apply this AAV$_{MUSE}$ to treat two chronic inflammatory diseases: nonalcoholic fatty liver disease (NAFLD) and allergic asthma, showing that inhalation of muscone—after only one injection of AAV$_{MUSE}$—can achieve long-term controllable expression of therapeutic proteins (ΔhFGF21 or ΔmIL-4). Our odorant-molecule-controlled system can advance gene-based precision therapies for human diseases.

Chronic diseases are long-lasting conditions with persistent effects and are a significant challenge for healthcare systems[1]. At present, therapeutic protein-based therapies, including antibodies and cytokines, are becoming widespread for treating chronic diseases; challenges with these macromolecular medications include the need for frequent parenteral administration by injections into skin, muscle, or veins, and high-dose treatment, which brings unwanted side effects such as allergic reactions and skin rashes[2,3]. There is an urgent need to develop safe and long-term therapeutic strategies for treating chronic diseases[4–6].

Owing to low immunogenicity and lack of pathogenicity, recombinant vectors based on adeno-associated viruses (AAVs) are gaining popularity as a platform for gene delivery to treat a variety of human diseases[7], and they have been successfully applied in both preclinical and clinical settings[8]. Moreover, AAV-based gene therapies can provide long-term therapeutic protein expression, which can obviate the need to manufacture and frequently inject short-lived recombinant proteins[9]. Recently, therapeutic genes have been delivered to specific, desired tissues by manipulating different serotypes

[1]Shanghai Frontiers Science Center of Genome Editing and Cell Therapy, Biomedical Synthetic Biology Research Center, Shanghai Key Laboratory of Regulatory Biology, Institute of Biomedical Sciences and School of Life Sciences, East China Normal University, Dongchuan Road 500, Shanghai 200241, China. [2]Institute of Medical Technology, Shanxi Medical University, Taiyuan, Shanxi Province 030001, China. [3]Chongqing Key Laboratory of Precision Optics, Chongqing Institute of East China Normal University, Chongqing 401120, China. [4]Department of Breast Surgery, Tongji Hospital, School of Medicine, Tongji University, Xincun Road 389, Shanghai 200065, China. [5]Wuhu Hospital, Health Science Center, East China Normal University, Middle Jiuhua Road 263, Wuhu, Anhui, China. [6]Division of Endocrinology, Department of Medicine, Sam and Ann Barshop Institute for Longevity and Aging Studies, University of Texas Health Science Center at San Antonio, San Antonio, TX 78229, USA. [7]These authors contributed equally: Xin Wu, Yuanhuan Yu, Meiyan Wang. ✉e-mail: hfye@bio.ecnu.edu.cn

and engineering AAV capsids[10–12]. However, the widespread adoption of AAV-based gene therapy technologies is limited partly by constitutive gene expression, which can lead to toxicity from supraphysiological levels of the therapeutic agent protein(s). For instance, AAV2/9-delivered *survival motor neuron* overexpression results in sensorimotor toxicity and reverses the initial benefits for spinal muscular atrophy therapy[13], and AAV-delivered *fibroblast growth factor 21* (*FGF21*) results in bone loss owing to constitutive gene expression at physiological concentrations[14]. Accordingly, there is a growing awareness that high and persistent transgene expression levels lead to side effects, and it is necessary to develop AAV vector technologies that enable the induction of gene expression within a suitable therapeutic window[15–17].

Towards this goal, various chemically-inducible systems have been incorporated into AAV vectors to control gene expression, for example, upon antibiotic exposure (e.g., tetracycline (Tet)[18] and erythromycin[19]). However, these trigger molecules are not suitable for controlling gene expression, because the use of antibiotics may promote antibiotic resistance and cause undesirable effects on animals, including cytotoxicity and tissue invasiveness[20,21]. The aroma compound, such as muscone secreted from male musk deer, has attracted interest for many years due to its fascinating and pleasant odor used in

perfumes and its physiological effects on humans[22]. Therefore, muscone is an ideal inducer for inducible systems to achieve needle-free aromatherapy.

Here, we develop a muscone-induced transgene system (MUSE) by taking advantage of a volatile and traceless muscone trigger. We package it into AAV vectors (hereafter referred to as "AAV_{MUSE}") to achieve controllable expression of therapeutic genes through inhalation of the muscone trigger, which provides needle-free aromatherapy. The AAV_{MUSE} system enables remote, muscone dose- and exposure-time-dependent induction of AAV-delivered transgene expression with a negligible background without the stimulus. We show repeated transgene activation that is inducible over 20 weeks after a single AAV administration. We also demonstrate that an AAV_{MUSE-ΔhFGF21} system using the liver tropism serotype AAV2/9 can temporally regulate the production of a therapeutic protein (ΔhFGF21) to treat nonalcoholic fatty liver disease (NAFLD) model mice and show that a MUSE iteration employing a combination of AAV serotypes (AAV2/6 and AAV2/lung) can control production of the therapeutic protein (ΔmIL-4) to treat allergic asthma model mice (Fig. 1a, b). Our study thus illustrates that AAV_{MUSE} is a valuable addition to the toolbox of controlled gene therapies that can improve safety and efficacy and advance gene therapy applications in precision medicine.

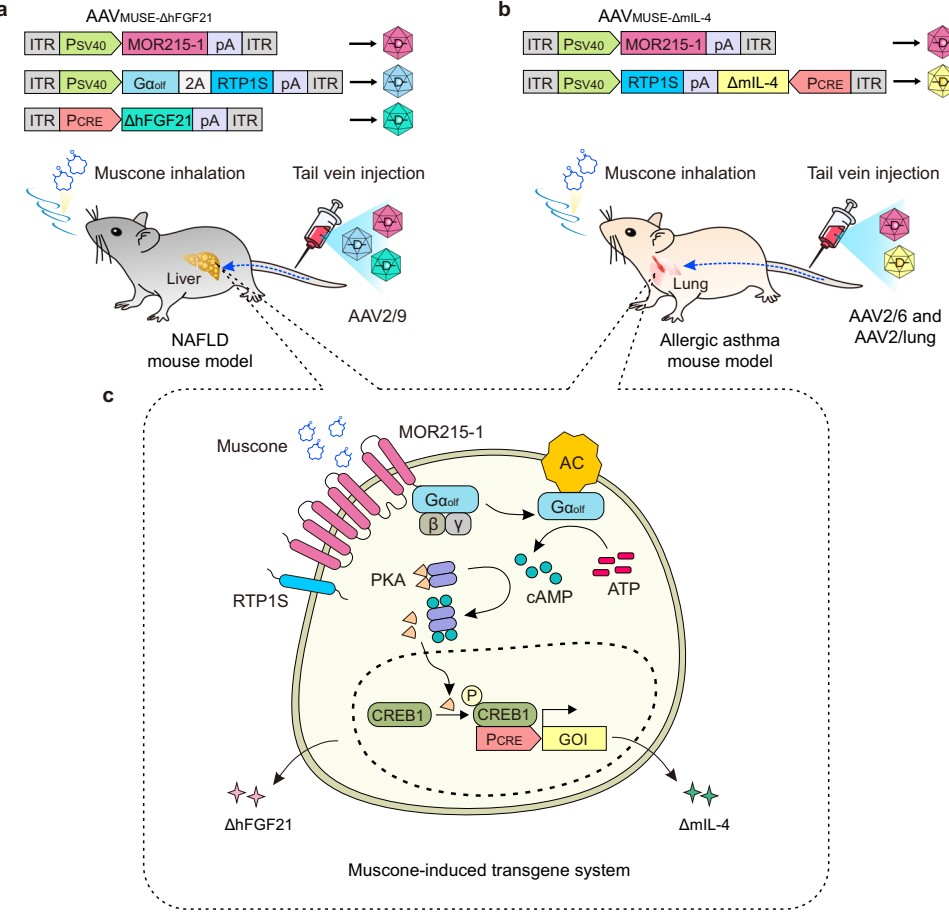

**Fig. 1 | Schematic showing the mouse experimental design for AAV_{MUSE}-mediated gene therapy to treat chronic diseases. a**, **b** Schematic representations of the genetic configurations for AAV_{MUSE-ΔhFGF21} and AAV_{MUSE-ΔmIL-4}. AAV_{MUSE-ΔhFGF21} containing the muscone-responsive vector AAV2/9-pWX126 (ITR-P_{SV40}-MOR215-1-pA-ITR), the concatenated Gα_{olf} and RTP1S expression vector AAV2/9-pWX127 (ITR-P_{SV40}-Gα_{olf}-P2A-RTP1S-pA-ITR), and the ΔhFGF21 inducible expression vector AAV2/9-pWX252 (ITR-P_{CRE}-ΔhFGF21-pA-ITR) were injected into diet-induced NAFLD model mice; AAV_{MUSE-ΔmIL-4} comprising AAV2/lung-pWX126 and AAV2/6-pWX345 (ITR-P_{SV40}-RTP1S::pA-ΔmIL-4-P_{CRE}-ITR) were injected into OVA-induced

allergic asthma model mice. **c** Detailed schematic for the muscone-induced transgene system (MUSE) design. The binding of muscone to the G-protein-coupled murine olfactory receptor (MOR215-1) triggers an intracellular surge of the second messenger cAMP via olfactory type G protein α subunit (Gα_{olf})-mediated activation of adenylate cyclase (AC). Then cAMP binds to the regulatory subunits of protein kinase A (PKA), the catalytic subunits of which are translocated into the nucleus, where they phosphorylate the cAMP-responsive binding protein 1 (CREB1), which binds to a synthetic cAMP-responsive promoter (P_{CRE}) to initiate transgene expression (e.g., ΔhFGF21 or ΔmIL-4).

## Results

### Design and functional validation of MUSE

To generate a muscone-induced transgene system (MUSE), we constitutively expressed the muscone-responsive murine olfactory receptor (MOR215-1) protein[23] and a truncated version of receptor-transporting protein 1 (RTP1S), which is known to facilitate MOR215-1 trafficking to the cell surface membrane[24]. When exposed to muscone, muscone binds to the G-protein-coupled receptor MOR215-1 to trigger an intracellular surge of the second messenger cAMP via $G\alpha_{olf}$-mediated activation of adenylate cyclase (AC)[25]. cAMP can then bind to the regulatory subunits of protein kinase A (PKA), the catalytic subunits of which translocate into the nucleus, where they phosphorylate the cAMP-responsive binding protein 1 (CREB1), which binds to a synthetic cAMP-responsive promoter ($P_{CRE}$) to initiate transgene expression (Fig. 1c).

To test whether MUSE can induce gene expression, we transfected human embryonic kidney cells (HEK-293T) with different combinations of plasmids encoding the muscone-responsive vector pMOR215-1, the cAMP-responsive promoter ($P_{CRE}$) driven reporter gene vector pCK53, a truncated version of the receptor-transporting protein vector pRTP1S, and an olfactory neuron-specific G protein alpha subunit vector $pG\alpha_{olf}$. The combination containing pMOR215-1, pRTP1S, $pG\alpha_{olf}$, and pCK53 had high induction (28-fold induction) of secreted human placental alkaline phosphatase (SEAP) expression in the presence of muscone in the culture medium (Supplementary Fig. 1a). To further improve reporter induction efficiency, we tested different ratios of plasmids encoding MUSE and found that a combination of pMOR215-1, pRTP1S, $pG\alpha_{olf}$, and pCK53 at a 5:1:1:1 ratio (w/w/w/w) showed the highest induction (63-fold) of SEAP expression in the presence of muscone in the culture medium (Supplementary Fig. 1b). Importantly, MOR215-1 could only be activated by muscone structural analogs (Supplementary Fig. 2) and was completely insensitive to the endogenously produced molecules that might cross-activate the cAMP signals (Supplementary Fig. 3). We also confirmed that MUSE was functional in additional mammalian cell lines (e.g., HeLa, hMSC-TERT, and Hana 3 A), suggesting its broad applicability (Fig. 2a).

To assess the possible cytotoxicity of muscone on cells, HEK-293T cells transfected with a constitutive SEAP expression vector (pSEAP2-control) were exposed to different concentrations of muscone (0–80 μM). SEAP expression was not reduced, indicating that

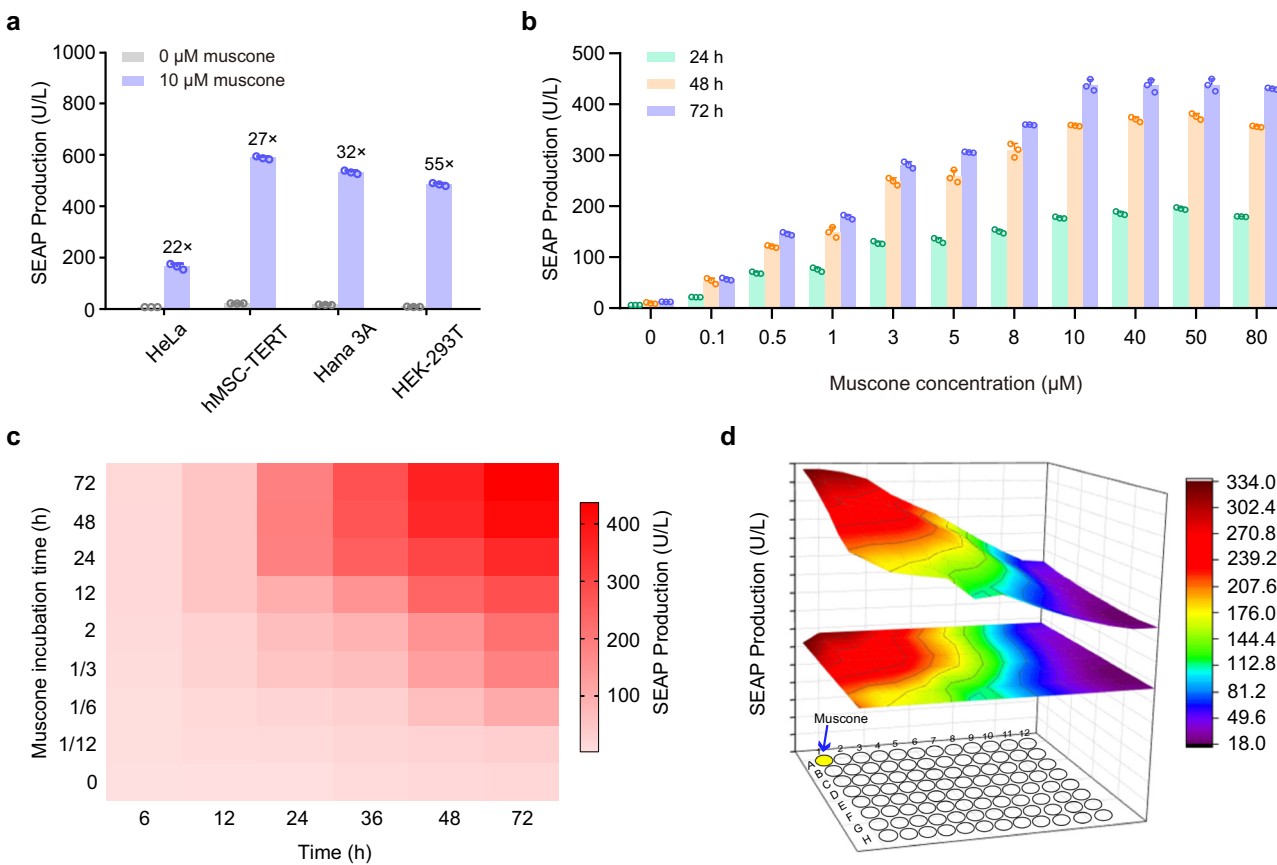

**Fig. 2 | Characterization of muscone-induced transgene system (MUSE) performance. a** Muscone-inducible SEAP expression in various mammalian cell lines. HeLa, hMSC-TERT, Hana 3 A, and HEK-293T cells were co-transfected with MUSE-encoding plasmids including pMOR215-1 ($P_{SV40}$-MOR215-1-pA), pRTP1S ($P_{SV40}$-RTP1S-pA), $pG\alpha_{olf}$ ($P_{SV40}$-$G\alpha_{olf}$-pA), and pCK53 reporter ($P_{CRE}$-SEAP-pA). Six hours after transfection, cells were cultivated in a culture medium with or without 10 μM muscone for 48 h, and SEAP production in the culture supernatant was profiled. **b** Muscone dose-dependent MUSE-mediated SEAP expression kinetics. pMOR215-1/pRTP1S/$pG\alpha_{olf}$/pCK53-cotransfected HEK-293T cells were cultivated in a cell culture medium containing the indicated concentrations of muscone, and SEAP expression was subsequently profiled every 24 h. **c** Muscone time-dependent MUSE-mediated SEAP expression kinetics. pMOR215-1/pRTP1S/$pG\alpha_{olf}$/pCK53-cotransfected HEK-293T cells were cultivated for the indicated periods (0 to 72 h) in the presence of

10 μM muscone. After the incubation, the cells were transferred into a muscone-free medium, and SEAP expression in the culture supernatant was profiled at the indicated time points (X-axis, 6 to 72 h). **d** Distance-dependent SEAP expression. MUSE-cotransfected HEK-293T cells seeded in a 96-well tissue culture plate were exposed to diluted muscone essential oil (1:100, v/v) and placed in the top left well that had no physical contact with the cultivated cells, and SEAP expression was profiled after 48 h. The three-dimensional surface plot indicates corresponding model-based simulations and represents numeric values for SEAP activity as per the indicated color scheme. Data in (**a**, **b**) are presented as means ± SD; $n = 3$ biologically independent samples. Descriptions of all plasmids, and detailed descriptions of the genetic constructs are provided in Supplementary Tables 1 and 3. Source data are provided as a Source Data file.

muscone exposure did not influence gene expression capacity (Supplementary Fig. 4a). Moreover, according to the cell viability assay, we did not observe any apparent toxicity for cells containing MUSE exposed to 0–80 μM muscone (Supplementary Fig. 4b), indicating that the MUSE components and 10 μM muscone are well tolerated by cells. Importantly, we found no apparent crosstalk between the MUSE-mediated cAMP signaling pathway and three other signaling pathways, including mitogen-activated protein kinase (MAPK)-mediated protein kinase (IRS-1-Ras-MAPK) pathway through the activation of insulin receptor (IR)[26], intracellular calcium-dependent nuclear factor of activated T cells (NFAT) pathway through the activation of transient receptor potential (TRP) melastatin 8 (TRPM8)[27], and proinflammatory nuclear factor NF-κB activation in response to toll-like receptor 2 (TLR2) signaling[28] (Supplementary Fig. 5a–i).

To evaluate MUSE-mediated SEAP expression kinetics, HEK-293T cells were co-transfected with MUSE-encoding plasmids and exposed to different concentrations of muscone (0–80 μM); SEAP expression was profiled every 24 h. SEAP expression was induced in a dose-dependent manner upon adjusting the muscone concentration (ranging from 0.1 to 10 μM) and achieved saturation above 10 μM muscone (Fig. 2b). HEK-293T cells co-transfected with MUSE-encoding plasmids were exposed to 10 μM muscone for various time periods; the anticipated time-dependent SEAP expression was detected (Fig. 2c). To characterize the remote responsiveness of MUSE-engineered cells to muscone, 10 μM muscone was placed at the upper left corner well of a 96-well plate; this induced distance-dependent SEAP expression in MUSE-engineered cells that were seeded throughout the other wells of the plate (Fig. 2d). Thus, our designed MUSE enabled safe and remote control of transgene expression in a muscone dose- and time-dependent manner in mammalian cells.

## AAV_MUSE-mediated transgene expression in mouse livers

Before transducing mice with MUSE using AAV2/9 delivery (AAV_MUSE), we developed a relatively simple system comprising a few constructs to facilitate in vivo delivery based on a single AAV vector that concatenated the constructs for RTP1S and $G\alpha_{olf}$; or MOR215-1 and RTP1S; or MOR215-1 and $G\alpha_{olf}$. We eventually obtained an optimized concatenated $G\alpha_{olf}$ and RTP1S vector pWX127 (ITR-$P_{SV40}$-$G\alpha_{olf}$-P2A-RTP1S-pA-ITR). We found that combining the muscone-responsive vector pWX126 (ITR-$P_{SV40}$-MOR215-1-pA-ITR), the concatenated $G\alpha_{olf}$ and RTP1S vector pWX127, and the reporter vector pWX158 (ITR-$P_{CRE}$-luciferase-P2A-EGFP-pA-ITR) at a 5:1:1 (w/w/w) ratio resulted in about 67-fold induction of luciferase expression (Supplementary Fig. 6). Further, AAV_MUSE-transfected HEK-293T cells were induced with muscone in a time-dependent manner, and our results indicated that the AAV_MUSE system exhibited fast kinetics, with significant luciferase expression within 2 h (Supplementary Fig. 7).

After validating the AAV_MUSE-mediated reporter gene expression in cells, we tested whether AAV_MUSE induced luciferase reporter expression upon exposure to nebulized-muscone using an ultrasonic nebulizer (Fig. 3a). Then mice were transduced with AAV2/9 vectors[29] carrying the AAV_MUSE system components via tail vein injection. At 2 weeks after the AAV injection, AAV_MUSE-transduced mice exposed to nebulized muscone showed high bioluminescence signal intensities in livers measured using an in vivo imaging system as compared to control mice transduced with AAV_MUSE and exposed to vehicle (a mixture of castor oil and ddH₂O) (Supplementary Fig. 8). We also observed higher bioluminescence signal in the isolated livers compared to other organs, indicating that the AAV2/9 efficiently delivered AAV_MUSE system into the liver after intravenous injection (Supplementary Fig. 9). It might be that liver cells are susceptible to AAV2/9 infection, allowing co-transfection of the three vectors in a single liver cell. Moreover, we measured the muscone concentration in the mouse liver, lung, blood, and in the ultrasonic nebulizer by gas

chromatography-mass spectrometry (GC-MS). Our results indicated that the concentration in these regions was within the range of the reported EC50 value for MOR215-1[30] (Supplementary Fig. 10).

To further characterize the kinetics of AAV_MUSE-mediated luciferase expression, AAV_MUSE-transduced mice were exposed to nebulized muscone and bioluminescence imaging was monitored every two hours (Supplementary Fig. 11a). We found that AAV_MUSE-transduced mice exposure to nebulized muscone for 2 h were sufficient to trigger luciferase expression and the maximum luciferase density was observed at 4 h after muscone exposure (Supplementary Fig. 11b, c). Moreover, AAV_MUSE-mediated luciferase expression was muscone dose- and exposure time-dependent in mice (Fig. 3b, c and Supplementary Fig. 12a, b). These results demonstrate that AAV_MUSE can efficiently induce luciferase expression in mice.

We next tested whether AAV_MUSE re-induced luciferase reporter expression upon re-exposure to nebulized muscone in mice. AAV_MUSE-transduced mice were exposed to nebulized muscone (18 mM) for 4 h once every two weeks; control mice were transduced with AAV_MUSE and exposed to vehicle biweekly. Based on weekly monitoring of bioluminescence imaging, we found that repeated muscone nebulization resulted in re-induction of luciferase expression in the livers of AAV_MUSE transduced mice (Fig. 3d and Supplementary Fig. 12c); mice given AAV_MUSE and exposed repeatedly to muscone displayed reproducible induction of luciferase expression for 28 weeks (Fig. 3e, f and Supplementary Fig. 13). However, AAV_MUSE-induced luciferase expression decreased with time, possibly due to rAAVs as non-replicating circularized episomes, gradually leading to the loss of transduced vectors in mitotic cells after AAV transduction[7]. In addition, the introduction of foreign proteins delivered by rAAVs into a mammalian organism may trigger adaptive or innate immune responses, potentially resulting in a reduction of transgene products[10,31–33].

Additional analyses from this long-term study to track potential side effects evident in routine blood test and serum biochemistry analyses showed no differences between AAV_MUSE-transduced mice exposed to muscone and control mice (including AAV_MUSE-transduced mice exposed to vehicle or un-transduced wild-type mice) (Supplementary Fig. 14). In addition, liver tissue samples were collected for Gene Set Enrichment Analysis (GSEA) after a total muscone induction period of 20 weeks. GSEA showed that other potential pathways might not be influenced by cAMP generated by the MUSE system in the livers of AAV_MUSE-transduced mice exposed to muscone as compared to control mice (including AAV_MUSE-transduced mice exposed to vehicle or un-transduced wild-type mice) (Supplementary Fig. 15), indicating that intermittent and transient stimulation of the MOR215-1 had not detrimental effects on the endogenous target genes of the cAMP signaling pathway due to transient nature of cAMP signaling activation[34,35]. Collectively, these results indicate that AAV_MUSE enables safe, remote, muscone dose- and time-dependent induction of transgene expression, and is re-inducible in mouse livers for up to 28 weeks upon muscone-mediated aromatherapy.

## Therapeutic efficacy of AAV_MUSE-ΔhFGF21 in diet-induced NAFLD model mice

Pursuing the idea of an AAV_MUSE-based gene therapy to treat NAFLD, we chose a synthetic recombinant Fc-FGF21 variant (ΔhFGF21) as the intervention[36,37]; this protein has been examined in a clinical Phase II trial designed to reduce hepatic fat in subjects with NAFLD[38]. To confirm the biological activity of AAV_MUSE-induced ΔhFGF21, we initially engineered a HEK-SEAP™ βKlotho/FGFR cell line co-transfected with three plasmids[39]: pβKlotho ($P_{CMV}$-βKlotho-pA), pTetR-ELK1 ($P_{CMV}$-TetR-ELK1-pA), and pMF111 ($P_{hCMV*-1}$-SEAP-pA; $P_{hCMV*-1}$, $O_{TetO7}$-$P_{hCMVmin}$) (Supplementary Fig. 16a). When ΔhFGF21 bound to FGFR1 and its coreceptor β-Klotho, the MAPK signaling pathway was activated to phosphorylate TetR-ELK1 (a synthetic hybrid transcription

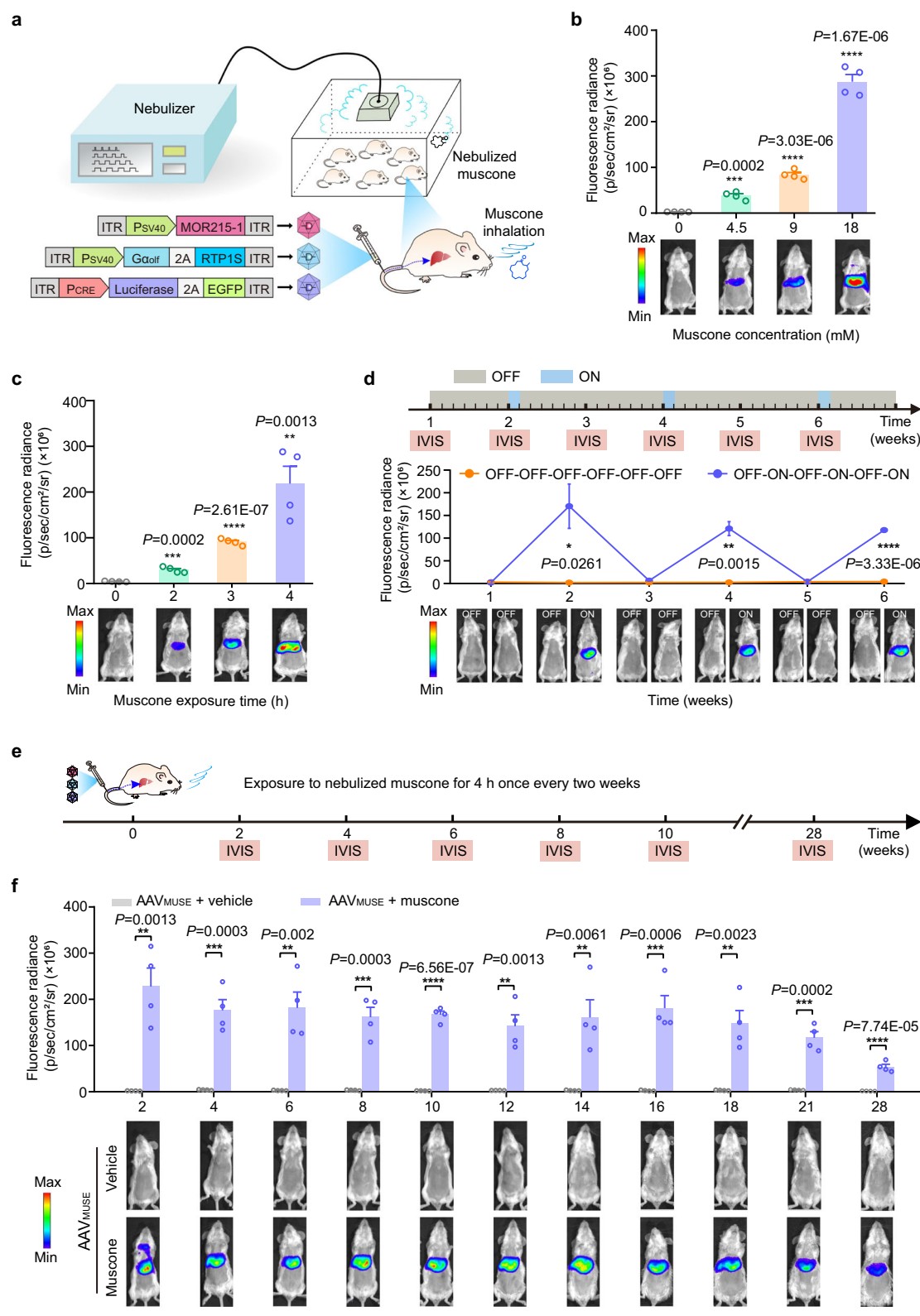

factor), leading to $P_{hCMV*-1}$-driven SEAP reporter expression (Supplementary Fig. 16b). Indeed, we observed that $AAV_{MUSE}$-induced ΔhFGF21 mediated time-dependent expression of SEAP in the engineered cells (Supplementary Fig. 16c).

After demonstrating the biological activity of ΔhFGF21, we evaluated the therapeutic efficacy of $AAV_{MUSE-ΔhFGF21}$-mediated gene therapy in diet-induced NAFLD model mice[40,41]. Ten weeks

after starting a high-fat, high-fructose, high-cholesterol diet (HFFCD), the induced NAFLD model mice were transduced with $AAV_{MUSE-ΔhFGF21}$ comprising the muscone-responsive vector AAV2/9-pWX126 ($1 \times 10^{11}$ viral genomes (vg)), the concatenated $Gα_{olf}$ and RTP1S vector AAV2/9-pWX127 ($5 \times 10^{10}$ vg), and the ΔhFGF21 vector AAV2/9-pWX252 (ITR-$P_{CRE}$-ΔhFGF21-pA-ITR, $5 \times 10^{10}$ vg) via tail vein (Fig. 4a and Supplementary Fig. 17a). After one week, the transduced NAFLD mice were

**Fig. 3 | AAV$_{MUSE}$-mediated luciferase reporter expression in mouse livers.**
**a** Schematic representation of the experimental procedure for AAV$_{MUSE}$-mediated luciferase expression in mice. Female BALB/c mice (8-week-old) were transduced with AAV$_{MUSE}$ containing the muscone-responsive vector AAV2/9-pWX126 ($3 \times 10^{11}$ vg), the concatenated G$\alpha_{olf}$ and RTP1S expression vector AAV2/9-pWX127 ($1.5 \times 10^{11}$ vg), and the inducible reporter vector AAV2/9-pWX158 (ITR-P$_{CRE}$-lucifer-ase-P2A-EGFP-pA-ITR, $1.5 \times 10^{11}$ vg) via tail vein injection. Two weeks after the AAV injection, AAV$_{MUSE}$-transduced mice were exposed to nebulized muscone using an ultrasonic nebulizer. **b** Muscone dose-dependent AAV$_{MUSE}$-mediated luciferase expression kinetics in mice. BALB/c mice transduced with AAV$_{MUSE}$ were exposed to nebulized muscone at the indicated concentrations for 4 h, and luciferase reporter expression was measured based on bioluminescence imaging 4 h after exposure to nebulized muscone. vg, viral genomes. **c** Muscone exposure time-dependent AAV$_{MUSE}$-mediated luciferase expression kinetics in mice. BALB/c mice transduced with AAV$_{MUSE}$ were exposed to nebulized muscone for the indicated time periods (0 to 4 h). **d** Re-inducibility of AAV$_{MUSE}$-mediated transgene expression in mice. According to the indicated time schedule, BALB/c mice transduced with AAV$_{MUSE}$ were exposed to nebulized muscone for 4 h once every two weeks; control mice were transduced with AAV$_{MUSE}$ without exposure to muscone. Bioluminescence imaging was monitored weekly using an in vivo imaging system 4 h after exposure to nebulized muscone. **e** Schematic representation of the schedule for AAV$_{MUSE}$-mediated luciferase expression in mice for 28 weeks. **f** BALB/c mice transduced with AAV$_{MUSE}$ were exposed to nebulized muscone or vehicle for 4 h once every two weeks. Bioluminescence imaging was performed at the indicated time points. Data in (**b**, **c**, **d** and **f**) are presented as means ± SEM (**b**, **c** and **f**, $n = 4$ mice; **d**, $n = 3$ mice). $P$ values were obtained from two-tailed unpaired $t$ tests. *$P < 0.05$, **$P < 0.01$, ***$P < 0.001$, ****$P < 0.0001$. Detailed images of the biolumi-nescence signals are provided in Supplementary Figs. 12 and 13. Source data are provided as a Source Data file.

exposed to nebulized muscone for 4 h once weekly until week 31 or were exposed to vehicle (a mixture of castor oil and ddH$_2$O). The AAV$_{MUSE-\Delta hFGF21}$-transduced NAFLD mice exposed to nebulized mus-cone (18 mM) exhibited a significant increase in ΔhFGF21 levels com-pared to vehicle-exposed control mice transduced with AAV$_{MUSE}$ over 21 weeks (Fig. 4b). Moreover, the AAV$_{MUSE}$-induced ΔhFGF21 expres-sion level reached a range of 20–30 ng/ml, which is within the desired therapeutic window.

At 19 or 29 weeks after starting the HFFCD diet, magnetic reso-nance imaging showed that AAV$_{MUSE-\Delta hFGF21}$-transduced NAFLD mice had significantly decreased fat mass percentage values and increased lean mass percentage values compared to control groups (including AAV$_{MUSE-\Delta hFGF21}$-transduced NAFLD mice exposed to vehicle, and AAV$_{Luc}$-transduced NAFLD mice with or without exposure to muscone) (Fig. 4c, d and Supplementary Fig. 17b, c). To assess the effects of AAV$_{MUSE-\Delta hFGF21}$ on metabolism, we placed the mice in indirect calori-metry cages at 20 or 27 weeks after starting the HFFCD diet and found that NAFLD mice transduced with AAV$_{MUSE-\Delta hFGF21}$ and exposed to muscone had significantly elevated oxygen consumption and carbon dioxide expiration (Supplementary Fig. 17d–g and Fig. 4e–h) com-pared to the relevant control mice at both time points. A battery of metabolic tests measuring plasma levels of liver enzymes, including aspartate aminotransferase (AST) and alanine aminotransferase (ALT), as well as total cholesterol (T-CHO) and triglyceride (TG) (Supple-mentary Fig. 17h–o), and an intraperitoneal glucose tolerance test (IPGTT) (Supplementary Fig. 18), consistently showed significant decreases in the AAV$_{MUSE-\Delta hFGF21}$-transduced NAFLD mice exposed to muscone as compared to the relevant control mice. Moreover, the AAV$_{MUSE-\Delta hFGF21}$-transduced NAFLD mice exposed to muscone had significantly reduced body weight compared to the appropriate con-trol mice (Fig. 4i, j).

Next, livers and epididymal white adipose tissue (eWAT) of mice were collected at the pre-determined end-point of 31 weeks. The AAV$_{MUSE-\Delta hFGF21}$-transduced NAFLD mice exposed to muscone dis-played significant decreases in the weight of livers and eWAT com-pared to the relevant control mice (Fig. 4k–n). Hematoxylin and eosin (H&E) and Sirius red staining of liver sections revealed that AAV$_{MUSE-\Delta hFGF21}$-transduced NAFLD mice exposed to muscone exhibited smaller adipose droplets, decreased severity of hepatic steatosis and hepatic fibrosis in the livers as compared to the relevant control mice (Fig. 4o). Taken together, these results confirm that AAV$_{MUSE-\Delta hFGF21}$ achieves re-induction of therapeutic protein at an appropriate level for treating NAFLD in diet-induced model mice after only one injection of AAVs.

## AAV$_{MUSE}$-mediated transgene expression in mouse lungs
To test whether AAV$_{MUSE}$ enables induction of transgene expression in mouse lungs, we initially used AAV2/lung[42] to package AAV$_{MUSE}$ containing three AAV vectors: pWX126 (ITR-P$_{SV40}$-MOR215-1-pA-ITR), pWX127 (ITR-P$_{SV40}$-G$\alpha_{olf}$-P2A-RTP1S-pA-ITR), and pWX158 (ITR-P$_{CRE}$-luciferase-P2A-EGFP-pA-ITR). Upon tail vein injection of varying titers of these vectors to BALB/c mice, no strong luciferase signal was detected in the lung (Supplementary Fig. 19). We speculated that the number of AAV$_{MUSE}$ component vectors or the capsid of the AAV vectors might have affected gene delivery efficiency.

Seeking to facilitate in vivo delivery, we first simplified AAV$_{MUSE}$ into two separate AAV vector plasmids: a muscone-responsive vector pWX126 and a concatenated vector for two expression cassettes (pWX322, ITR-P$_{SV40}$-RTP1S-pA::P$_{CRE}$-SEAP-pA-ITR or pWX325, ITR-P$_{SV40}$-RTP1S::pA-SEAP-P$_{CRE}$-ITR) (Supplementary Fig. 20a, b). Com-pared to pWX126/pWX322, the pWX126/pWX325 combination exhibited a higher 21.5-fold induction of SEAP expression and lower background signal. We eventually selected the concatenated construct pWX325 (ITR-P$_{SV40}$-RTP1S::pA-SEAP-P$_{CRE}$-ITR). A combina-tion of pWX126 and the concatenated vector pWX325 at a 5:1 ratio (w/w) resulted in about 21.5-fold muscone induction of SEAP expres-sion in HEK-293 cells. We also detected about 16.9-fold muscone induction of a firefly luciferase reporter in engineered cells co-transfected with pWX126 and pWX342 (ITR-P$_{SV40}$-RTP1S::pA-lucifer-ase-P$_{CRE}$-ITR) (Supplementary Fig. 20c). Moreover, we selected AAV serotype combinations for delivering constitutive luciferase (ITR-P$_{hCMV}$-luciferase-pA-ITR) to the lung via the tail vein and demon-strated that the combination of AAV2/lung and AAV2/6 resulted in higher bioluminescence signal intensities in lungs compared to either AAV2/6 alone or AAV2/lung alone (Supplementary Fig. 21). We then used a combination of two AAV serotypes for delivery of AAV$_{MUSE}$: pWX126 packaged into a AAV2/lung capsid (which was reported to transduce the endothelium of the pulmonary vasculature efficiently) and pWX342 packaged into a AAV2/6 capsid[43] (which was reported to transduce both airway and alveolar type II cells efficiently). To test whether AAV$_{MUSE}$ induced luciferase reporter expression upon expo-sure to nebulized-muscone in the lung, adult BALB/c mice were transduced with AAV$_{MUSE}$ comprising AAV2/lung-pWX126 ($5 \times 10^{11}$ vg) and AAV2/6-pWX342 ($5 \times 10^{11}$ vg) via tail vein (Fig. 5a). Two weeks after AAV injection, AAV$_{MUSE}$-transduced mice were exposed to nebulized muscone for 4 h once every six weeks or exposed to vehicle (a mixture of castor oil and ddH$_2$O) (Fig. 5b). We found that AAV$_{MUSE}$-transduced mice exposed to nebulized muscone exhibited significantly higher bioluminescence signal intensities (up to 17-fold induction) in lungs compared to vehicle-exposed control mice transduced with AAV$_{MUSE}$. Moreover, bioluminescence imaging at different time points showed that the AAV$_{MUSE}$-transduced mice exposed to nebulized muscone expressed the luciferase reporter in lungs for up to 20 weeks (Fig. 5c, d). Collectively, these results indicate that the AAV$_{MUSE}$ enables long-term induction of transgene expression in the lungs when the animal is exposed regularly to the inducer.

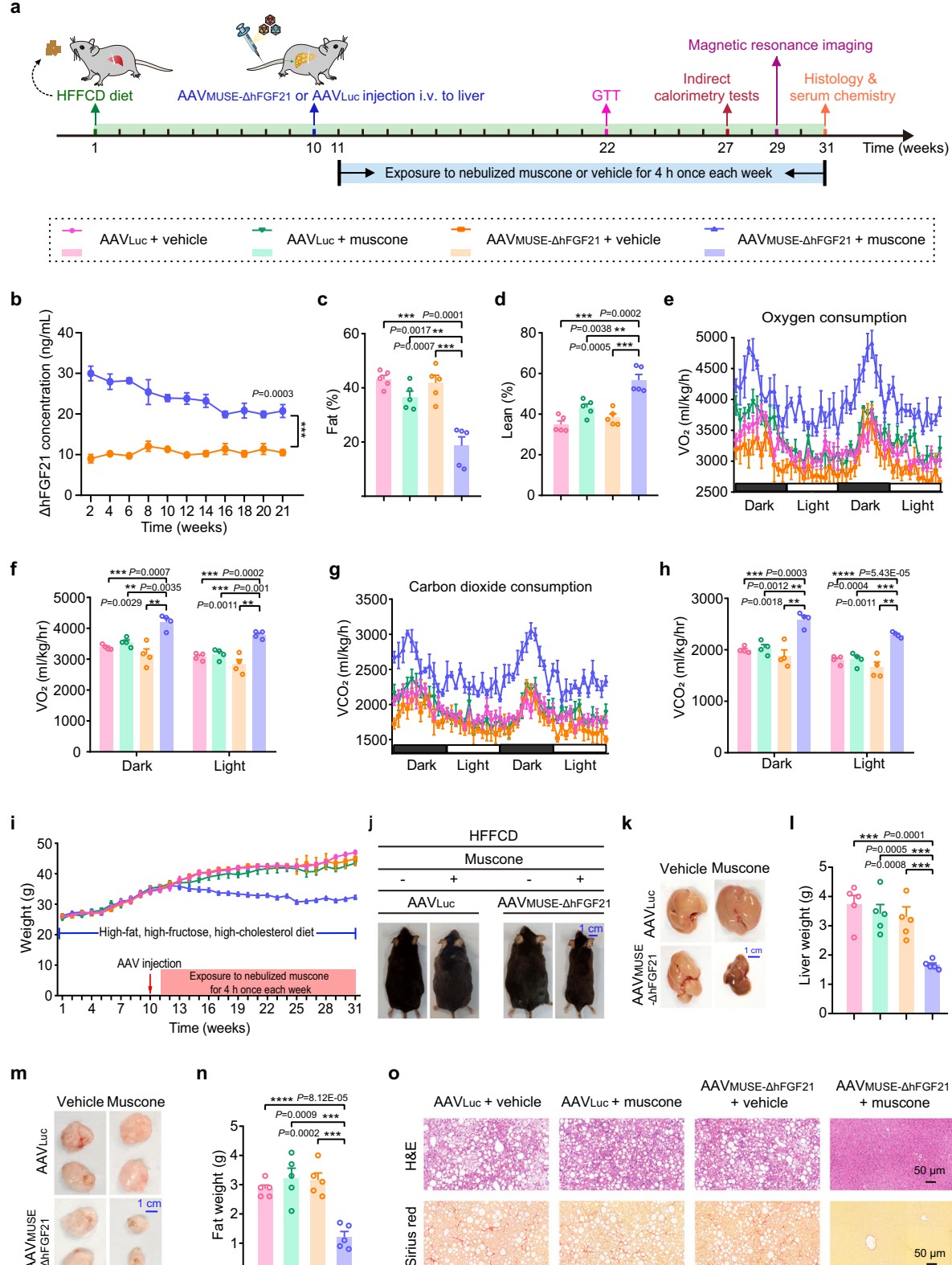

## Therapeutic efficacy of AAV$_{MUSE-\Delta mIL-4}$ in allergic asthma model mice

We next developed AAV$_{MUSE}$ for an in vivo gene therapy application to control the expression of a murine IL-4 mutant protein ($\Delta$mIL-4)[44], which has been recently reported to protect against allergic asthma[45]. To verify the biological activity of AAV$_{MUSE}$-induced $\Delta$mIL-4, we initially engineered HEK-SEAP™ IL-4R$\alpha$/IL-13R$\alpha$1 cells[46,47] by co-transfecting

plasmids including an interleukin-4 receptor $\alpha$ subunit plasmid pIL-4R$\alpha$ (P$_{CMV}$-IL-4R$\alpha$-pA), an interleukin-13 receptor $\alpha$1 subunit plasmid pIL-13R$\alpha$1 (P$_{CMV}$-IL-13R$\alpha$1-pA), a mouse signal transducer and activator of transcription 6 plasmid pSTAT6 (LTR-P$_{CMV}$-mSTAT6::P$_{PGK}$-puromycin-LTR), and a STAT6-inducible SEAP reporter plasmid pWX326 (P$_{STAT6}$-SEAP-pA; P$_{STAT6}$, O$_{STAT6}$-P$_{min}$) (Supplementary Fig. 22a). Consistent with the expected outcomes, when $\Delta$mIL-4 bound to the

**Fig. 4 | The therapeutic efficacy of AAV$_{MUSE-\Delta hFGF21}$-mediated gene therapy in diet-induced NAFLD model mice. a** Schematic representation of the experimental procedure and schedule for AAV$_{MUSE}$-mediated $\Delta$hFGF21 expression (AAV$_{MUSE-\Delta hFGF21}$) in NAFLD model mice. At 10 weeks after starting the HFFCD diet, NAFLD model mice were transduced with AAV$_{MUSE-\Delta hFGF21}$ or transduced with control AAV$_{Luc}$ via tail vein. One week after AAV injection, the transduced NAFLD model mice were exposed to nebulized muscone for 4 h once each week until week 31 or were exposed to a vehicle (a mixture of castor oil and ddH$_2$O). The HFFCD diet was continued for an additional 21 weeks, and a battery of metabolic parameters in mice were measured at the indicated time points. i.v., intravenous. **b** The $\Delta$hFGF21 levels in plasma were quantified 4 h after muscone exposure every two weeks over 21 weeks. **c, d** The fat mass and lean mass were measured by magnetic resonance imaging at 29 weeks after starting the HFFCD diet. **e−h** Indirect calorimetry was performed 27 weeks after starting the HFFCD diet in

NAFLD model mice. **e** Volume of O$_2$ (VO$_2$) consumption. **f** Area under the curve for O$_2$ consumption. **g** Volume of CO$_2$ (VCO$_2$) expiration. **h** Area under the curve for CO$_2$ expiration. **i** The body weight of NAFLD model mice were monitored every week over 31 weeks. **j** Representative images of the NAFLD model mice 31 weeks after starting the HFFCD diet. Representative pictures of dissected livers (**k**) and epididymal adipose tissues (**m**); the weight of liver tissues (**l**) and epididymal adipose tissues (**n**) were determined 31 weeks after starting the HFFCD diet. **o** Representative H&E is staining, and Sirius red staining of liver sections. The images represent typical results from three independent measurements. Scale bars, 50 μm. Data in (**b**−**i**, **l** and **n**) are presented as means ± SEM (**e**−**h**, $n$ = 4 mice; **b**−**d**, **i**, **l** and **n**, $n$ = 5 mice). $P$ values were obtained from two-tailed unpaired $t$ tests. *$P$ < 0.05, **$P$ < 0.01, ***$P$ < 0.001, ****$P$ < 0.0001. Source data are provided as a Source Data file.

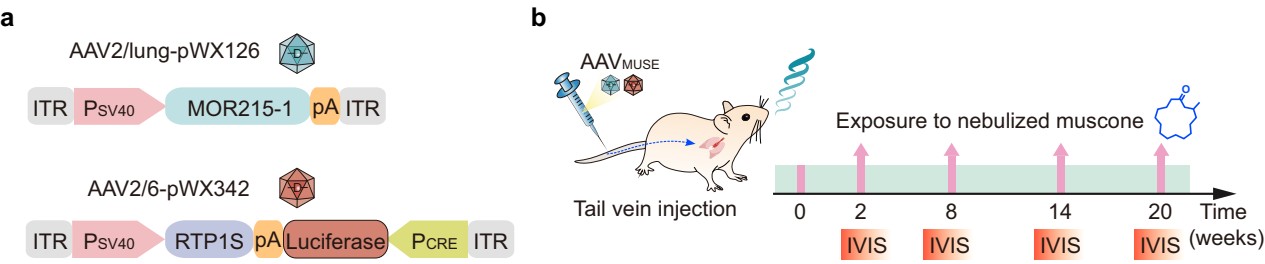

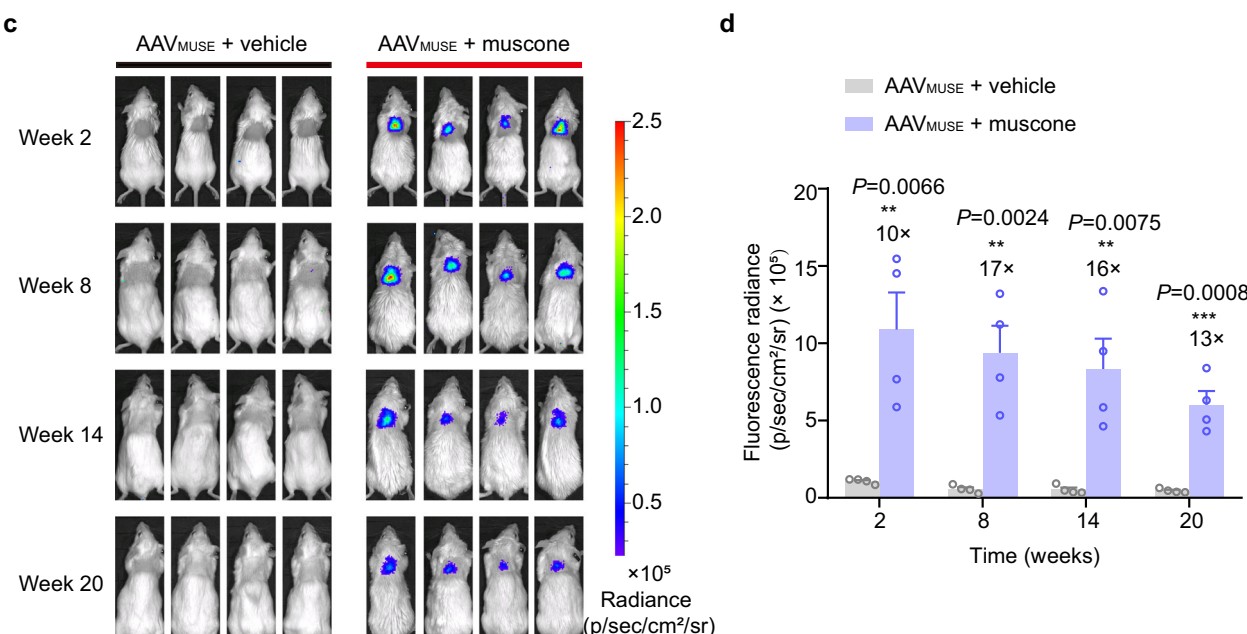

**Fig. 5 | AAV$_{MUSE}$-mediated luciferase reporter expression in mouse lungs. a** Schematic representing the genetic configuration of the AAV vector used for AAV$_{MUSE}$ iteration. **b** Schematic representation of the experimental procedure for AAV$_{MUSE}$-mediated transgene expression in mouse lungs. **c** AAV$_{MUSE}$-mediated luciferase expression for 20 weeks in mouse lungs. Eight-week-old female BALB/c mice were transduced with an AAV$_{MUSE}$ iteration comprising AAV2/lung-pWX126 (ITR-P$_{SV40}$-MOR215-1-pA-ITR, $5 \times 10^{11}$ vg) and AAV2/6-pWX342 (ITR-P$_{SV40}$-RTP1S::pA-luciferase-P$_{CRE}$-ITR, $5 \times 10^{11}$ vg) via tail vein. Two weeks after the AAV injection,

AAV$_{MUSE}$-transduced mice were exposed to nebulized muscone for 4 h once every six weeks or exposed to a vehicle (a mixture of castor oil and ddH$_2$O), and bioluminescence imaging was performed using an in vivo imaging system every 6 weeks. **d** Quantification of luciferase expression in the lung based on the bioluminescence IVIS imaging shown in (**c**). Data in (**d**) are presented as means ± SEM ($n$ = 4 mice). **$P$ < 0.01, ***$P$ < 0.001. $P$ values were obtained from two-tailed unpaired $t$ tests. Source data are provided as a Source Data file.

IL-4Rα/IL-13α1 heterodimeric receptor complex, IL-4/IL-13 signaling was suppressed, which inhibited P$_{STAT6}$-driven SEAP reporter expression (Supplementary Fig. 22b). We observed that AAV$_{MUSE}$-induced $\Delta$mIL-4 expression was muscone exposure time-dependent in these engineered cells (Supplementary Fig. 22c).

After confirming the biological activity of $\Delta$mIL-4, we evaluated the therapeutic efficacy of AAV$_{MUSE-\Delta mIL-4}$ mediated gene therapy in ovalbumin (OVA)-induced allergic asthma model mice[48,49]. Fourteen days before first sensitization, BALB/c mice were transduced with AAV$_{MUSE-\Delta mIL-4}$ (containing AAV2/lung-pWX126, $5 \times 10^{11}$ vg, and

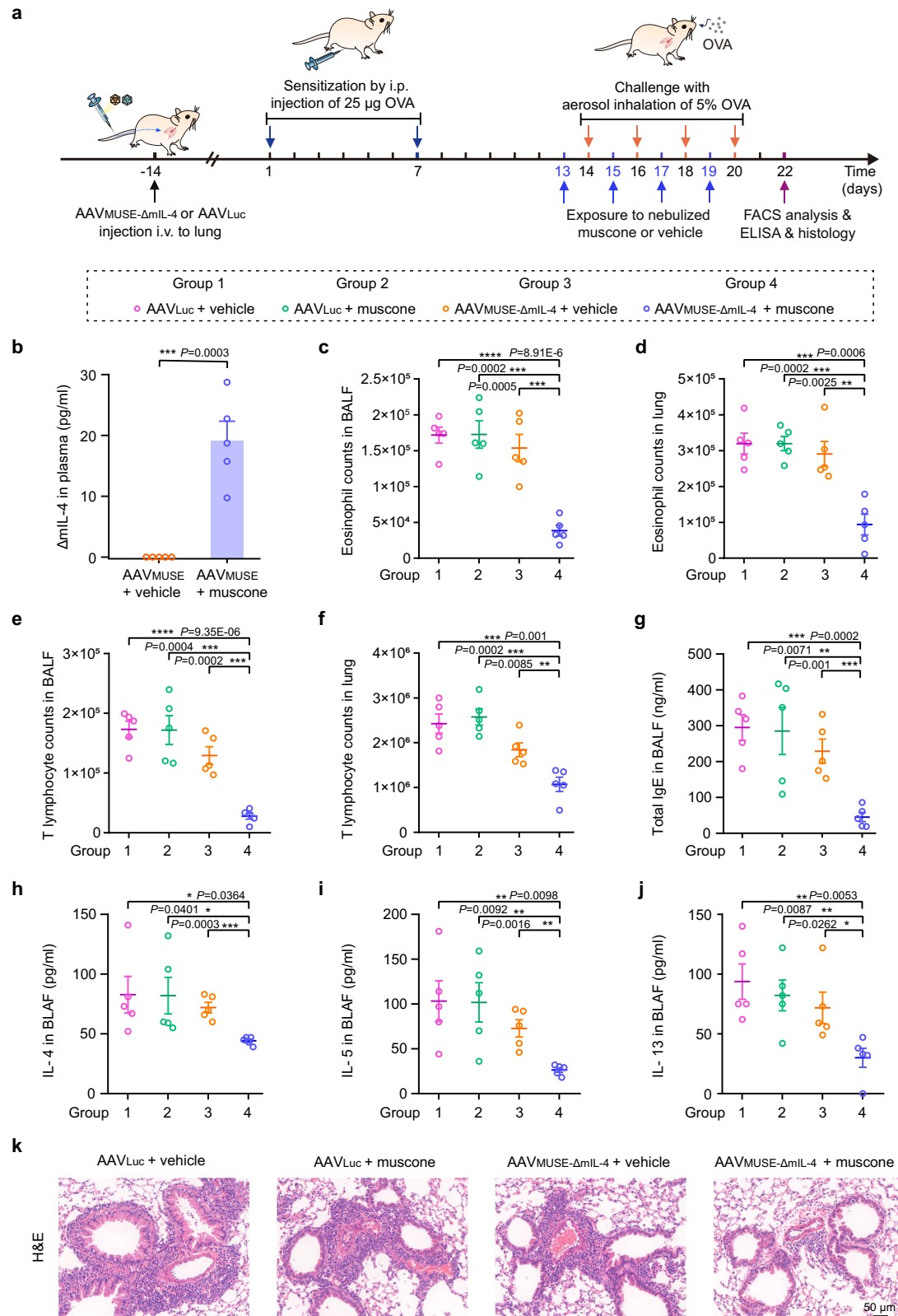

AAV2/6-pWX345, $5 \times 10^{11}$ vg) via tail vein injection; control mice were transduced only with the luciferase reporter AAV$_{Luc}$ ($1 \times 10^{12}$ vg). All mice were then sensitized by intraperitoneal (i.p.) injection of 25 µg OVA (Grade V) adjuvanted with aluminum hydroxide (Alum) on days 1 and 7, followed by challenge with aerosol inhalation of 5% OVA (Grade II) for 20 min on days 14, 16, 18, and 20 (Fig. 6a).

One day before the OVA challenge, the AAV$_{MUSE-ΔmIL-4}$-transduced mice or AAV$_{Luc}$-transduced control mice were exposed to nebulized muscone or vehicle for 4 h on days 13, 15, 17, and 19. We found that the AAV$_{MUSE-ΔmIL-4}$-transduced group exposed to muscone induced significantly higher ΔmIL-4 levels compared to AAV$_{MUSE-ΔmIL-4}$-transduced control mice exposed to vehicle (Fig. 6b). On day 22, lungs and

**Fig. 6 | The therapeutic efficacy of AAV$_{MUSE-\Delta mIL-4}$ mediated gene therapy in the OVA-induced allergic asthma model mice. a** Schematic representation of the experimental procedure and time schedule for AAV$_{MUSE}$-mediated $\Delta mIL$-4 expression (AAV$_{MUSE-\Delta mIL-4}$) in allergic asthma model mice. Fourteen days before the first sensitization, BALB/c mice were transduced with AAV$_{MUSE-\Delta mIL-4}$ containing AAV2/lung-pWX126 (ITR-P$_{SV40}$-MOR215-1-pA-ITR, $5 \times 10^{11}$ vg) and AAV2/6-pWX345 (ITR-P$_{SV40}$-RTP1S::pA-$\Delta mIL$-4-P$_{CRE}$-ITR, $5 \times 10^{11}$ vg), and control mice were transduced with only the luciferase reporter AAV$_{Luc}$ ($1 \times 10^{12}$ vg) via tail vein injection. Then all mice were sensitized by intraperitoneal (i.p.) injection of 25 μg OVA (Grade V) on days 1 and 7, followed by challenge with aerosol inhalation of 5% OVA (Grade II) for 20 min on days 14, 16, 18, and 20 to induce the allergic asthma model. One day before the OVA challenge, the AAV$_{MUSE-\Delta mIL-4}$-transduced mice or AAV$_{Luc}$-transduced control mice were exposed to nebulized muscone or vehicle for 4 h every two days (on days 13, 15, 17, and 19; blue arrow). AAV$_{MUSE-\Delta mIL-4}$ transduced mice

exposed to a vehicle (a mixture of castor oil and ddH$_2$O), and AAV$_{Luc}$ transduced mice exposed to muscone or vehicle were used as controls. On day 22, lungs and bronchoalveolar lavage fluid (BALF) were collected. **b** The levels of $\Delta mIL$-4 in plasma were quantified using a mouse IL-4 ELISA kit. **c, d** The counts of eosinophils in lungs and BALF were measured by flow cytometry. **e, f** Lymphocytes were measured in the lungs and BALF by flow cytometry. **g** The total mouse IgE in BALF was measured using a mouse IgE ELISA kit. **h–j** The pro-inflammatory Th2-type cytokines including IL-4, IL-5, and IL-13 were quantified in BALF using mouse ELISA kits. **k** Representative images of hematoxylin and eosin (H&E) staining of lung sections. The images represent typical results from three independent measurements. Scale bars, 50 μm. Data in (**b–j**) are presented as means ± SEM (*n* = 5 mice per group). *$P < 0.05$, **$P < 0.01$, ***$P < 0.001$, ****$P < 0.0001$. *P* values were obtained from two-tailed unpaired *t* tests. Source data are provided as a Source Data file.

bronchoalveolar lavage fluid (BALF) were collected and flow cytometry analysis showed that AAV$_{MUSE-\Delta mIL-4}$-transduced mice exposed to muscone had significantly reduced counts of eosinophils (CD11b$^+$siglec-F$^+$) and T lymphocytes (CD3$^+$) in both BALF and lungs as compared to control groups (including AAV$_{MUSE-\Delta mIL-4}$-transduced allergic asthma model mice exposed to vehicle, AAV$_{Luc}$-transduced allergic asthma mice with or without exposure to muscone) (Fig. 6c–f and Supplementary Fig. 23).

Moreover, the levels of Th2-type pro-inflammatory cytokines (IL-4, IL-5, IL-13) and total IgE levels were significantly decreased in the BALF of AAV$_{MUSE-\Delta mIL-4}$-transduced mice exposed to muscone as compared to the relevant controls (Fig. 6g–j). Additionally, H&E staining of lung sections revealed that AAV$_{MUSE-\Delta mIL-4}$-transduced mice exposed to muscone exhibited significantly decreased eosinophil infiltration surrounding airways as compared to the relevant controls (Fig. 6k). Together, these results demonstrate that AAV$_{MUSE-\Delta mIL-4}$ can remotely control $\Delta mIL$-4 to inhibit eosinophil infiltration and pro-inflammatory cytokines and to prevent the development of sensitivity to OVA in allergic asthma mice based on inhalation of muscone.

## Discussion

AAV vectors are a leading platform for gene delivery to treat a variety of human diseases[7]. However, clinical experience with multiple AAV-based gene therapies has raised awareness that the supraphysiological levels of transgene expression might induce potential adverse effects[15]. Therefore, inducible gene expression systems are attractive approaches for controlling the expression of AAV-delivered genes. Here, we developed AAV$_{MUSE}$ for remote induction of AAV-delivered transgene expression in a muscone dose- and exposure-time-dependent manner. We demonstrate that AAV$_{MUSE}$ iterations can be efficiently transduced into mouse livers or lungs, and can induce reporter gene expression for at least 20 weeks after a single AAV administration when mice are exposed repeatedly to nebulized muscone. Importantly, we have also demonstrated that AAV$_{MUSE}$-based gene therapy achieved dynamic regulation of therapeutic protein production ($\Delta hFGF21$ in liver and $\Delta mIL$-4 in lung) at physiologically appropriate levels in NAFLD and allergic asthma mice models through one-time intravenous delivery of AAV$_{MUSE}$ followed by exposure to muscone.

AAV$_{MUSE}$ is based on a muscone trigger; it employs needle-free aromatherapy[50], a convenient alternative to injection for chronic conditions such as diabetes, obesity, and chronic kidney disease. Moreover, our AAV$_{MUSE}$ can be delivered directly to tissues of interest and achieve long-term, controllable production of therapeutic proteins after only one injection of AAV vectors. These aspects are very attractive for therapy applications requiring sustained gene expression compared to cell-based therapies, which require cell collection and ex vivo culture[51]. Moreover, some therapeutic cells might be encapsulated inside a semipermeable and immune isolating container to enhance xenogeneic cell survival[52], making it challenging to achieve long-term gene expression.

MUSE system components can be delivered into various target tissues by engineering the AAV capsid with enhanced tissue tropism, for example, by using directed evolution and through different modes of administration[53]. It may be possible to deliver MUSE to the brain using the brain tropism serotype AAV-PHP.eB[54] to control production of a therapeutic protein (e.g., methyl-CpG binding protein 2, MECP2) for treating Rett syndrome (RTT), which could avoid promotion of neurological disorders known to result from aberrant overexpression of MECP2[55]. MUSE could also be used to regulate the production of erythropoietin (EPO) for treating anemia associated with chronic kidney disease[56] using the engineered muscle tropism serotype MyoAAV after systemic administration[57], which could avoid the induction of high plasma EPO levels linked to elevated risk for cardiovascular complications[58]. AAV$_{MUSE}$ could also be transduced into liver tissue to serve as a bio-factory to control the production of various therapeutic proteins based on controlling muscone dose and exposure time compared to traditional drugs that are static and cannot be adjusted, including thymic stromal lymphopoietin (TSLP)[59] for treating obesity and FGF21 for treating diabetes[60].

Despite our successful demonstrations of treating chronic diseases using AAV$_{MUSE}$-based gene therapies, some barriers must be overcome before chemically inducible systems can be widely accepted for a broad range of diseases, such as the potential cytotoxicity from the chemical inducers that affect cell numbers and colony formation[61], and the stability and metabolism of the chemical inducer that affects the duration and consistency of gene expression or diffusing freely in vivo[62]. Moreover, multiple components of our system are an obstacle to broad application. Possible solutions include designing a smaller version of MUSE suitable for packaging within a single AAV by truncating and mutating the AAV$_{MUSE}$ modules containing RTP1S and MOR215-1, perhaps based on artificial intelligence (AI)-guided protein structure prediction and classification[63,64] or via high-throughput ligand screening[65]. Another idea for the efficient in vivo delivery of MUSE is using engineered virus-like particle (eVLP)[66], which hold great promise as vehicles for targeted therapeutic macromolecule delivery and represent an improvement over AAV or plasmid delivery. Further, for the tissue specific delivery of our AAV$_{MUSE}$ system to livers, it can be resolved by employing liver-specific promoters to drive MOR215 and utilizing engineered tissue-specific AAV capsids through directed evolution and computer-guided design[7]. We anticipate that our designed AAV$_{MUSE}$ system for controlling therapeutic gene expression will be a valuable addition to gene-based therapeutics and can help develop more effective and safer personalized disease treatments.

## Methods

### Ethical statement

This study did not involve any human data. Experiments involving animals were performed according to a protocol approved by the East China Normal University (ECNU) Animal Care and Use Committee and in accordance with the Ministry of Science and Technology of the

People's Republic of China on Animal Care guidelines. The protocols (protocol ID: m20200213, m20210114) were approved by the ECNU Animal Care and Use Committee. All animals were euthanized after the termination of the experiments.

## Animals

Female BALB/c mice (8-week-old) and male C57BL/6 mice (8-week-old) were obtained from the ECNU (East China Normal University) Laboratory Animal Center. The mice were housed at the ECNU Animal Care in 12/12-h light/dark cycles at $22 \pm 2\,°C$ and the humidity from 40% to 60%. The 3–5 mice were housed per cage in individually ventilated cages, with ad libitum access to food and water. The mouse gender was taken into account in the study design. For the NAFLD model, male C57BL/6 mice (8-week-old) are commonly chosen to construct the NAFLD model due to their susceptibility to diet-induced obesity and metabolic complications[67]. For the allergic asthma model, female BALB/c mice (8-week-old) are selected to construct the allergic asthma model because female mice have been reported to develop a more pronounced type of allergic airway inflammation than male mice after the OVA challenge[48]. The corresponding figure legends state the sex, age, strain, and number of mice used for each experiment. At the end of the study, mice were euthanized by carbon dioxide exposure.

## Plasmid construction

Plasmids constructed and used in this study are provided in Supplementary Table 1. The detailed DNA sequences for MUSE components are provided in Supplementary Table 3. Some plasmids were constructed using ClonExpress Ultra One Step Cloning kit (C116, Vazyme Biotech Co., Ltd) according to the manufacturer's instructions. All genetic components were confirmed by Sanger sequencing (Shanghai Saiheng Biotechnology). The fragments for ΔhFGF21 [the fusion protein of hFc and hFGF21 (GenBank accession number NC_000019.10) with three mutations (L98R, P171G, and A180E)] and ΔmIL-4 [IL-4 (GenBank accession number NC_000077.7) with two mutations (R116D, Y119D)] were chemically synthesized by the Genewiz company.

## Cell culture and transfection

Human embryonic kidney cells (HEK-293T, CRL-11268, ATCC); human cervical adenocarcinoma cells (HeLa, CCL-2, ATCC); HEK-293-derived Hana3A cells engineered for constitutive expression of RTP1, RTP2, REEP1, and $G_{\alpha o \lambda \phi}$[68]; and telomerase-immortalized human mesenchymal stem cells (hMSC-TERT, SCRC-4000, ATCC) were cultured in Dulbecco's modified Eagle's medium (DMEM, catalog no. C11995500BT, Gibco) supplemented with 10% (v/v) fetal bovine serum (FBS, catalog no. FBSSA500-S, AusGeneX) and 1% (v/v) penicillin/streptomycin solution (catalog no. ST488-1/ST488-2, Beyotime). All cells were cultured at $37\,°C$ in a humidified atmosphere containing 5% $CO_2$ and regularly tested for the absence of Mycoplasma and bacterial contamination.

Cells were transfected with an optimized polyethyleneimine (PEI)-based protocol. Briefly, cells were seeded into a 48-well plate ($3 \times 10^4$ cells per well) and cultivated overnight to 60–70% confluency at the time of transfection. Cells were subsequently co-transfected with corresponding plasmid mixtures for 6 h with 50 μL of PEI and DNA mixture (PEI and DNA at a mass ratio of 3:1 for HEK-293T, hMSC-TERT, and Hana3A, and at a mass ratio of 5:1 for HeLa; PEI, molecular weight 40,000, stock solution 1 mg/mL in double distilled water, catalog no. 24765, Polysciences). At 6 h after transfection, the culture medium was replaced with fresh medium.

## Chemicals

Odorants utilized in this study were purchased from Macklin Biochemical Technology Corporation. Castor oil was purchased from Aladdin. The musk odorants and musk analogs were prepared as 200 mM stock solutions in dimethyl sulfoxide or anhydrous ethanol.

Menthol (catalog no. HY-75161), dopamine (catalog no. HY-B0451A), and CU-T12-9 (catalog no. HY-110353) were purchased from MedChemExpress (China) and freshly dissolved in water before each experiment. GLP-1 (catalog no. G3265), atrial natriuretic peptide (ANP) (catalog no. A1663), and human insulin solution (catalog no. I9278) were purchased from Sigma-Aldrich (Buchs, Switzerland).

## SEAP reporter assay

The expression of human placental SEAP in a cell culture medium was quantified using a p-nitrophenyl phosphate-based light absorbance time course assay. Briefly, 120 μL of substrate solution [100 μL of 2 × SEAP buffer (pH 9.8) containing 20 mM L-homoarginine hydrochloride (catalog no. A602842, Sangon Biotech), 1 mM $MgCl_2$ (catalog no. A610328, Sangon Biotech), and 21% (v/v) diethanolamine (catalog no. A600162, Sangon Biotech), and 20 μL of substrate solution containing 120 mM p-nitrophenyl phosphate (catalog no. 333338-18-4, Sangon Biotech)] were added to 80 μL of heat-inactivated ($65\,°C$, 30 min) cell culture supernatant[69]. The time course of absorbance at 405 nm was measured at $37\,°C$ using a Synergy H1 hybrid multimode microplate reader (BioTek Instruments) with Gen5 software (version 2.04).

## Luciferase assay

HEK-293T cells ($6 \times 10^4$) were transfected with the AAV$_{MUSE}$ system (pWX126, pWX127, pWX158) and cultured for indicated time periods (0 to 24 h) in the presence or absence of 10 μM muscone. Then, cells were collected immediately at the indicated time points, and luciferase activity was measured using the Firefly Luciferase Reporter Gene Assay Kit (catalog no. RG005, Shanghai Beyotime Biotechnology Co., Ltd.). The time course of absorbance at 510 nm was measured at $37\,°C$ using a Synergy H1 hybrid multimode microplate reader (BioTek Instruments) with Gen5 software (version 2.04).

## Quantitative real-time PCR

Tissue samples from the left lateral lobe of the liver were harvested, snap-frozen in liquid nitrogen, and stored at $-80\,°C$ until use. Total RNA was extracted using an RNAiso Plus kit (catalog no. 9109; Takara Bio), and RNA concentration was measured using NanoDrop OneC (ThermoFisher). A total of 1 μg of RNA was reversely transcribed into cDNA using a HiScript® III RT SuperMix with the genomic DNA Eraser (catalog no. R323, Vazyme). Real-time PCR was performed on a Real-Time PCR Instrument (Roche, LightCycler 96, Switzerland) with ChamQ Universal SYBR qPCR Master Mix (catalog no. Q711, Vazyme). The following parameters were used for the PCR: $95\,°C$ for 10 min followed by 40 cycles at $95\,°C$ for 30 s, $55\,°C$ for 30 s, and $72\,°C$ for 30 s, and a final extension at $72\,°C$ for 10 min. All samples were normalized to the housekeeping gene glyceraldehyde 3-phosphate dehydrogenase (GAPDH). Values and the results were expressed as a relative mRNA amount using the standard $2^{-\Delta\Delta Ct}$ method. The sequences of primer pairs used in the study are listed in Supplementary Table 2.

## RNA-sequencing

Livers were dissected from AAV$_{MUSE}$-transduced mice exposed to muscone and control mice (including AAV$_{MUSE}$-transduced mice exposed to vehicle or un-transduced wild-type mice), and resuspended in TRIzol reagent (catalog no. 15596026, Invitrogen). Total RNA was extracted and treated with DNase I (Qiagen, Dusseldorf). Library preparation, quality control, and sequencing were performed by Genechem (Shanghai, China). RNA sequencing was performed via the Illumina platform (Illumina Novaseq 6000, CA) based on the mechanism of SBS (sequencing by synthesis). Gene set enrichment analysis (GSEA) was performed on a local tool http://www.broadinstitute.org/gsea/index.jsp. The raw data of RNA-seq was uploaded into the NCBI Sequencing Read Archive (SRA) under accession number PRJNA1011482.

### Complete blood count and analysis of liver function

Mice were sacrificed, and whole blood was collected and immediately analyzed for complete blood count using the Sysmex XT-2000iV hematology analyzer (Sysmex). Indicators of hepatic function, including alanine aminotransferase (ALT) and aspartate aminotransferase (AST), were measured using an automatic biochemical analyzer BX-3010 (Sysmex). All biochemical serum evaluations were performed simultaneously to minimize analytical variability.

### AAV production

AAV (serotype 2/9) carrying the muscone-responsive G-protein-coupled receptor MOR215-1 (pWX126, ITR-$P_{SV40}$-MOR215-1-pA-ITR), a concatenated $G\alpha_{olf}$ and RTP1S expression vector (pWX127, ITR-$P_{SV40}$-$G\alpha_{olf}$-P2A-RTP1S-pA-ITR), the inducible luciferase reporter vector (pWX158, ITR-$P_{CRE}$-luciferase-P2A-EGFP-pA-ITR), or the inducible ΔhFGF21 expression vector (pWX252, ITR-$P_{CRE}$-ΔhFGF21-pA-ITR) were produced by Shanghai Taitool Bioscience. Purified individual AAVs were titered using a quantitative PCR and concentrated in phosphate-buffered saline (PBS, 10 mM, pH 7.4) to $2.15 \times 10^{13}$, $2 \times 10^{13}$, $1.73 \times 10^{13}$, $2.23 \times 10^{13}$ viral genomes (vg)/mL.

AAV (serotype 2/6) carrying pWX342 (ITR-$P_{SV40}$-RTP1S::pA-luciferase-$P_{CRE}$-ITR) or pWX345 (ITR-$P_{SV40}$-RTP1S::pA-ΔmIL-4-$P_{CRE}$-ITR) and AAV (serotype 2/lung) carrying pWX126, pWX127, or pWX158 were produced by Shanghai OBiO Technology. Purified individual AAVs were titered using a quantitative PCR and concentrated in PBS to $2 \times 10^{13}$, $2 \times 10^{13}$, $1.8 \times 10^{13}$, $1.4 \times 10^{13}$, or $1.8 \times 10^{13}$ vg/mL.

### Mice exposed to nebulized muscone using an ultrasonic nebulizer

Adult female BALB/c mice (8-week-old) or male C57BL/6 mice (8-week-old, East China Normal University) were maintained in an appropriate climatized environment, with controlled conditions of temperature ($22 \pm 2$ °C), humidity (55%), and photoperiod (12 h of light/12 h of dark) for at least 7 days before starting all studies. Mice were transduced with AAV2/9 vectors carrying the AAV$_{MUSE}$ components via the tail vein. At 2 weeks after AAV injection, the AAV$_{MUSE}$-transduced mice were placed into a $285 \times 240 \times 160$ mm chamber and exposed to nebulized muscone [63 µL muscone (catalog no. B21154, Shanghai yuanye Bio-Technology) dissolved in a mixture of 700 µL castor oil (catalog no. C107105, Aladdin) and 3.5 mL ddH$_2$O] using an ultrasonic nebulizer (SA703, Jiangsu SANS Biological Technology) for different time periods (0–4 h). Four hours after muscone inhalation, bioluminescence images of the mice were obtained using an IVIS Lumina II in vivo imaging system (Perkin Elmer), and analyzed with Living Image software (version 4.3.1).

### Instruments and mass spectrometric conditions

The quantitative analysis was performed on an Agilent 7890B gas chromatography system coupled with an Agilent 7000D triple quadrupole mass spectrometer (Agilent Technologies, USA). The processed samples were placed in the vials of the autosampler for injection. The chromatographic column was the DB-5UI capillary column (30 m × 0.25 mm, 0.25 µm). The separation procedure of the GC system was set at the initial temperature of 80 °C held for 1 min, then heated up to 300 °C at the speed of 10 °C/min and held for 5 min. No splitting mode was set. Helium was the carrier gas with a 1.0 mL/min flow rate. The temperature of interface, ion source, and Quadrupole temperature were set at 280 °C, 230 °C, and 150 °C, respectively.

### Preparation of standard and sample solutions

The stock solution of muscone (20 µg/mL) was prepared in methanol and diluted with ethyl acetate to produce 0–1000 ng/mL working standard solution used in the linearity experiment. The internal standard (IS) working solution (30 µg/mL) was prepared by diluting Naphthalene stock solutions (2 mg/mL) with ethyl acetate. The

Calibration standards were prepared by spiking standard working solution (10 µL) and IS working solution (10 µL) with blank plasma (90 µL) to obtain a serial of muscone concentrations (0, 1, 3, 5, 10, 30, and 100 ng/mL). The sample treatment was a simple liquid-liquid extraction (LLE). The 0.1 mL mouse blood was collected from the orbital venous plexus and transferred to 1.5 mL EDTA-polythene tubes at 4 h after exposure to nebulized muscone. The plasma samples were prepared by spiking 50 µL plasma, 10 µL IS, 50 µL blank plasma with 200 µL neat ethyl acetate. The livers and lungs were dissected from mice 4 h after exposure to nebulized muscone. The liver and lung samples were prepared by spiking 100 µL sample, 10 µL IS with 200 µL neat ethyl acetate. The muscone concentration in the ultrasonic nebulizer's headspace was analyzed after diluting by 1000 times, and then 100 µL diluted muscone was added with 10 µL IS and 200 µL ethyl acetate. Then, the mixture solution was vortexed for 1.0 min and centrifuged at $13,000 \times g$ at 4 °C for 10 min. The supernatant was transferred into an injection vial for analysis. All the prepared solutions were stored at −20 °C until analysis. 1 µL supernatant was subjected to GC-MS/MS system for analysis.

### AAV$_{MUSE}$-mediated luciferase expression in mouse livers

Wild-type BALB/c mice (8-weeks-old) were transduced with AAV$_{MUSE}$ containing the muscone-responsive vector AAV2/9-pWX126 ($3 \times 10^{11}$ vg), the concatenated $G\alpha_{olf}$ and RTP1S expression vector AAV2/9-pWX127 ($1.5 \times 10^{11}$ vg), and the inducible reporter expression vector AAV2/9-pWX158 (ITR-$P_{CRE}$-luciferase-P2A-EGFP-pA-ITR, $1.5 \times 10^{11}$ vg) via tail vein. Two weeks after AAV injection, the AAV$_{MUSE}$-transduced mice were exposed to nebulized muscone using a nebulizer for different time periods (0–4 h) and different muscone concentrations (0–18 mM); control mice were transduced with AAV$_{MUSE}$ and exposed to vehicle (a mixture of castor oil and ddH$_2$O). Four hours after muscone inhalation, each mouse was intraperitoneally injected with luciferin (150 mg kg$^{-1}$; catalog no. luc001, Shanghai Sciencelight Biology Science & Technology) under anesthesia. Five minutes after luciferin injection, bioluminescence images of the mice were obtained using an IVIS Lumina II in vivo imaging system (Perkin Elmer), and analyzed with Living Image software (version 4.3.1).

### AAV$_{MUSE}$-mediated ΔhFGF21 production in diet-induced NAFLD model mice

Eight-week-old male C57BL/6 mice were fed a high-fat, high-fructose, high-cholesterol diet (HFFCD, 40% kcal fat, 20% kcal fructose, and 2% cholesterol; catalog no. D09100301, Research Diet) to induce non-alcoholic fatty liver disease (NAFLD)[40,41]. At 10 weeks after starting the HFFCD diet, NAFLD model mice were transduced with AAV$_{MUSE-\Delta hFGF21}$ containing the muscone-responsive receptor expression vector AAV2/9-pWX126 ($1 \times 10^{11}$ vg), the concatenated $G\alpha_{olf}$ and RTP1S expression vector AAV2/9-pWX127 ($5 \times 10^{10}$ vg), and the inducible ΔhFGF21 expression vector AAV2/9-pWX252 (ITR-$P_{CRE}$-ΔhFGF21-pA-ITR, $5 \times 10^{10}$ vg), or transduced with firefly luciferase control vector AAV$_{Luc}$ ($2 \times 10^{11}$ vg; ITR-$P_{CBh}$-mCherry-P2A-luciferase-pA-ITR) via tail vein. One week after AAV injection, the transduced NAFLD model mice were exposed to nebulized muscone or vehicle (a mixture of castor oil and ddH$_2$O) for 4 h once each week until week 31. The examined controls included NAFLD model mice transduced with AAV$_{MUSE-\Delta hFGF21}$ and exposed to a vehicle, as well as NAFLD model mice transduced with AAV$_{Luc}$ with or without exposure to muscone. All groups of mice were maintained on the HFFCD diet during the study. After 4 h of exposure to muscone, blood samples were collected from mice orbits and transferred to ethylenediaminetetraacetic acid (EDTA) coated mini vacutainer tubes (catalog no. 36597499, BD). After 10 minutes of incubation, samples were centrifuged at $1000 \times g$ for 10 min at 4 °C to separate plasma. ΔhFGF21 levels in mouse plasma were quantified using a human FGF21 ELISA kit (EK1151, MultiSciences Biotech) every 2 weeks from 10 to 31 weeks after starting the HFFCD diet. Body

weights were monitored weekly for 31 weeks, and magnetic resonance imaging was performed to measure the fat mass and lean mass at 19 or 29 weeks after starting the HFFCD diet using AccuFat nuclear magnetic resonance (NMR) (AccuFat-1050, MAG-MED).

## Measurement of triglycerides (TG) and total cholesterol (T-CHO) levels in mice plasma

At 21 or 31 weeks after starting the HFFCD diet, mouse peripheral blood samples were collected from mice orbits and transferred to EDTA-coated mini vacutainer tubes (catalog no. 36597499, BD). After 10 min of incubation, blood samples were centrifuged at $1000 \times g$ for 10 min at 4 °C. Plasma was separated and stored at −80 °C for further analysis. The levels of triglycerides (TG) and total cholesterol (T-CHO) in mouse plasma were measured using commercial kits [632-50991 (TG) and 294-65801 (T-CHO), Wako] according to the manufacturer's instructions, and tests were performed with the Synergy H1 hybrid multimode microplate reader (BioTek Instruments) with Gen5 software (version 2.04).

## Intraperitoneal glucose tolerance test (IPGTT) in NAFLD model mice

At 22 weeks after starting the HFFCD diet, all mice were fasted for 16 h and received an intraperitoneal injection of D-glucose (1.5 g/kg body weight) dissolved in PBS. The glycemic profile of each animal was monitored via tail vein blood samples at 0, 30, 60, 90, and 120 min after D-glucose administration using a Contour Glucometer (Exactive Easy III, MicroTech Medical). The trapezoidal rule determined the area under the curve (AUC) for IPGTT.

## Indirect calorimetry tests

At 20 or 27 weeks after starting the HFFCD diet, all mice were individually housed in metabolic cages of a Comprehensive Lab Animal Monitoring System (CLAMS, Columbus Instruments, Columbus) at 24 °C–26 °C. After a 24-h acclimation period, animals were monitored for 48 h to determine the calorimetric parameters for $VO_2$, $VCO_2$, and body weight. All mice were exposed to a daily 12-h light phase (07:00–19:00) and a 12-h dark phase (20:00–06:00).

## Ovalbumin (OVA)-induced allergic asthma model mice

To establish an OVA-induced allergic asthma mouse model, each female BALB/c mouse was sensitized by intraperitoneal injection of 25 μg OVA (Grade V, catalog no. A5503, Sigma-Aldrich) emulsified with 2 mg aluminum hydroxide (alum adjuvant, catalog no. 77161, Thermo Fisher Scientific) in 100 μL PBS on days 1 and 7. Mice were then challenged with nebulized 5% OVA [500 mg OVA (Grade II, catalog no. A5253, Sigma-Aldrich) dissolved in 10 mL PBS] for 20 min once at days 14, 16, 18, and 20. Control mice were similarly sensitized and challenged with PBS instead of OVA. To verify that the allergic asthma disease model was successfully established, bronchoalveolar lavage fluid (BALF) and lung tissues were harvested 48 h after the final challenge to assess eosinophil recruitment. Levels of IL-4, IL-5, and IL-13 in BALF were determined using corresponding ELISA kits according to the manufacturer's instructions.

## AAV$_{MUSE}$-mediated ΔmIL-4 production in OVA-induced allergic asthma model mice

Fourteen days before the first sensitization (day -14), female BALB/c mice (8-week-old) were transduced with AAV$_{MUSE-ΔmIL-4}$ containing AAV2/lung-pWX126 ($5 \times 10^{11}$ vg) and AAV2/6-pWX345 ($5 \times 10^{11}$ vg) via tail vein injection; control mice were transduced only with the luciferase reporter AAV$_{Luc}$ ($1 \times 10^{12}$ vg). All mice were then sensitized by intraperitoneal injection of 25 μg OVA (Grade V) adjuvanted with aluminum hydroxide (Alum) on days 1 and 7, followed by challenge with aerosol inhalation of 5% OVA (Grade II) for 20 min once on days 14, 16, 18, and 20. AAV$_{MUSE-ΔmIL-4}$-transduced mice or AAV$_{Luc}$-transduced

control mice were exposed to nebulized muscone or vehicle for 4 h on days 13, 15, 17, and 19. On day 22, mice were euthanized with carbon dioxide asphyxiation, and lungs and BALF were collected for further analysis.

## Bronchoalveolar lavage procedure

Mice were euthanized with carbon dioxide asphyxiation, and a small semi-incision of the trachea was made using scissors, allowing a 24-G lavage tube to pass into the trachea. The tube and trachea were stabilized by attaching them with a cotton thread, then a 1-mL syringe was placed into the 24-G lavage tube, and 0.5 mL sterile 1×PBS loaded in the syringe was injected into the lung, which was repeated three times. BALF was pooled and centrifuged for 5 min at $300 \times g$ at 4 °C to pellet the cells. The supernatant was separated and stored at -80 °C for further analysis of cytokines, and the cell pellets were resuspended in 300 μL RPMI 1640 medium for further flow cytometry analysis.

## Isolation of lung tissue cells

Mice were euthanized with carbon dioxide asphyxiation, and lung tissues were collected, cut into small pieces, and enzymatically digested in 5 mL solution containing 250 U/mL collagenase VIII (catalog no. C2139, Sigma-Aldrich) and 75 μg/mL DNase I (catalog no. DN25, Sigma-Aldrich) for 60 min at 37 °C with mild shaking. Lung cell suspensions were passed through a 70 μm cell strainer (strong nylon mesh with 70-micron pores, catalog no. 93070, Thermo Fisher Scientific) and centrifuged at $300 \times g$ for 5 min. Pelleted cells were resuspended in 4 mL of 30% Percoll® (catalog no. 17089109-1 EA, Cytiva), and 2.5 mL of 70% Percoll® was added to the bottom. Subsequently, the samples were centrifuged at $600 \times g$ with acceleration (ACE) = 5, deceleration (DCE) = 1 for 20 min, and the intermediate liquid containing the lung tissue cells were washed using 9 mL 1× PBS and then centrifuged at $800 \times g$ for 20 min to remove the supernatants. Finally, the cell pellets were resuspended in 200 μL RPMI 1640 medium (catalog no. 11875093, Gibco) on ice for flow cytometry analysis.

## Flow cytometry analysis of eosinophils and lymphocytes in BALF and lung tissues

To quantify the counts of eosinophils, cells from BALF or lung tissues were resuspended in 100 μL RPMI 1640 medium containing BV510 Live/Dead (catalog no. 423101, Biolegend), Alexa Fluor®-700 anti-mouse CD45 (catalog no. 103127, clone 30-F11, Biolegend), KIRAVIA Blue 520™ anti-mouse F4/80 (catalog no. 123161, clone BM8, Biolegend), Brilliant Violet 421™ anti-mouse CD11c (catalog no. 117329, clone N418, Biolegend), APC/Cyanine7-anti-mouse CD11b (catalog no. 101226, clone M1/70, Biolegend), and PE anti-mouse CD170 (Siglec-F) (catalog no. 155505, clone S17007L, Biolegend) on ice for 30 minutes. To quantify the counts of T lymphocytes, cells from BALF or lung tissues were resuspended in 100 μL RPMI 1640 medium containing BV510 Live/Dead (catalog no. 423101, Biolegend), FITC anti-mouse CD3ε (catalog no.100306, clone 145-2C11, Biolegend) on ice for 30 min. All cells were washed with PBS and were measured on a BD LSRFortessa™ Flow Cytometer (BD Biosciences). All antibodies were purchased from Biolegend and were diluted at 1:200. A minimum of 5000 events per plot were collected and analyzed using FlowJo V10 software. The numbers presented in the flow cytometry analysis images are percentage-based. All antibodies were validated for the specified application by respective manufacturer.

## Enzyme-linked immunosorbent assay (ELISA)

Levels of ΔhFGF21 in mouse plasma were quantified using a human FGF21 ELISA kit (EK1151, MultiSciences Biotech). Levels of IL-4, IL-5, or IL-13 in mouse BALF were quantified using a mouse IL-4 ELISA kit (abs520003, Absin), a mouse IL-5 High Sensitivity ELISA kit (EK205HS, MultiSciences Biotech), or a mouse IL-13 ELISA kit (KMC2221, Invitrogen), respectively. The levels of total IgE were quantified using a mouse

IgE ELISA kit (1218202, Biosci). All ELISA assays were conducted according to the manufacturer's instructions.

## Hematoxylin and eosin (H&E) staining of liver and lung tissues

Mice were sacrificed by carbon dioxide asphyxiation, and samples from liver and lung tissue were collected and fixed in 4% paraformaldehyde (catalog no. G1101, Servicebio) overnight at room temperature. The fixed samples were gently dehydrated by immersing in a graded series of alcohol solutions, cleaned in xylene, embedded in paraffin, and then sliced into 4 μm-thick tissue sections with a rotary microtome (Leica RM2235, Manual Rotary Microtome). These sections were stained with the H&E Staining Kit (catalog no. G1005, Servicebio) according to the manufacturer's instructions and were observed using an upright microscope (BX53, Olympus) equipped with an Olympus digital camera.

## Sirius red staining of liver tissues

Liver tissues were fixed overnight in 4% paraformaldehyde, embedded in paraffin, and cut into 4-μm sections. Liver sections were stained with 0.05% Sirius red solution (G1018, Servicebio) for 8 min, dehydrated quickly with anhydrous ethanol, cleaned in xylene for 5 min, and sealed with neutral gum. Representative Sirius red staining of liver sections was observed using an upright microscope (BX53, Olympus) equipped with an Olympus digital camera.

## Statistical analysis

All in vitro data are expressed as the mean ± SD of three independent biological replicates and are described separately in the figure legends. For the animal experiments, each treatment group consisted of randomly selected mice ($n = 3$ to 5), and the results are expressed as means ± SEM. Neither animals nor samples were excluded from the study. Comparisons between the two groups were performed using two-tailed unpaired $t$ tests. One-way ANOVA was used to evaluate differences between multiple groups with a single intervention, followed by Dunnett's post hoc test. Statistical analyses were performed using GraphPad Prism software (version 8), and differences were considered statistically significant at $*P < 0.05$, $**P < 0.01$, $***P < 0.001$, and $****P < 0.0001$. $n$ and $P$ values are described in the figures or figure legends.

## Reporting summary

Further information on research design is available in the Nature Portfolio Reporting Summary linked to this article.

# Data availability

All data associated with this study are present in the paper or the Supplementary Information. Source data are provided as a Source Data file with this paper. The raw data of RNA-seq was uploaded into the NCBI Sequencing Read Archive (SRA) under accession number PRJNA1011482. The remaining data are available within the Article, Supplementary Information or Source Data file. All genetic components related to this paper are available with a material transfer agreement and can be requested from H.Y. (hfye@bio.ecnu.edu.cn). Source data are provided with this paper.

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

## Acknowledgements

We thank Prof. Dr. Kazushige Touhara and Prof. Dr. Mika Shirasu (Department of Applied Biological Chemistry, The University of Tokyo) for providing cDNA for pMOR215-1, pRTP1S, and pGα$_{olf}$. We thank Hongyi Zang (Shanghai East Hospital, Tongji University School of Medicine) for assistance with animal experiments. This work was financially supported by grants from the National Key R&D Program of China, Synthetic Biology Research (no.2019YFA0904500), the National Natural Science Foundation of China (no. 32250010, and 32261160373), the Science and Technology Commission of Shanghai Municipality (no. 23HC1410100

and 22N31900300), and the Fundamental Research Funds for the Central Universities to H.Y. This work was also partially supported by National Key R&D Program of China (no. 2019YFA0110802), the National Natural Science Foundation of China (NSFC: no. 32171414), the Natural Science Foundation of Shanghai (no. 23ZR1419500), and the Nature Science Foundation of Chongqing, China (no. CSTB2022NSCQ-MSX0461) to M.W. This work was also partially supported by the Young Scientists Fund of the National Natural Science Foundation of China (no. 32301217) to J.Y. and the Young Scientists Fund of the National Natural Science Foundation of China (no. 32300458) to Y.Z. We thank the ECNU Multifunctional Platform for Innovation (011) for supporting the mouse studies and the Instruments Sharing Platform of School of Life Sciences, East China Normal University. We also thank the support from the Chinese Academy of Sciences Youth Interdisciplinary Team.

## Author contributions

H.Y. conceived of the project. H.Y., X.W., Y.Y. and W.L. designed the experiments and analyzed and interpreted the results. H.Y., X.W., Y.Y., M.W., J.Y. and S.Z. wrote the manuscript. X.W., Y.Y., M.W., W.L., J.Y., D.D., D.K., S.T., M.M., T.G., Y.Z., N.G. and Y.Z. performed the experimental work. All authors edited and approved the final manuscript.

## Competing interests

The authors declare no competing interests.
