## [Peer Review File · Nature Communications]

Reviewers' Comments:

Reviewer #1:

Remarks to the Author:

Taking advantage of a volatile non-invasive character of an odorant, the authors developed a system to express a specific olfactory receptor in a target tissue in the mouse body using adeno-associated virus vector and to stimulate the receptor by letting the infected mouse inhale the ligand odorant. This system successfully allowed for induction of a long-term controllable expression of a desired proteins associated with two chronic inflammatory diseases. This is a unique gene therapy system that could be widely used for a specifically targeted precision therapy for various human diseases.

1. Since mice express endogenously MOR215-1 in the nose, it should be confirmed that various therapeutic effects are via an AAV-delivered MOR215-1 expressed in liver or lung tissues but not via an olfactory tissue-expressed MOR215-1. The easiest way is to examine the effects in MOR215-1 null mice.
2. The muscone concentration in the headspace of an ultrasonic nebulizer should be measured. Another question in this line is how much muscone is delivered to liver and lung. You can extract the tissue by an organic solvent and measure the amount of muscone by GC-MS, or can measure the amount of muscone in the blood. The amount should be consistent with the EC50 value for MOR215-1.
3. To show that the induction of AAV-mediated transgene expression is indeed mediated by MOR215-1, non-ligand for MOR215-1, for example described in Sato-Akuhara et al. J. Neurosci. 2016, should be tested as a negative control. Having results in this experiment, one can also suggest that the use of an olfactory receptor is advantageous because of the high ligand specificity.
4. In the lung experiments, the response was seen without Golf, which is in contrast with the case of liver. Does this mean that the MOR215-1 is coupling with endogenous Gs? Then, why is Golf necessary in liver that should also express Gs? It has been known that the lung expresses Golf (Aisenberg, Sci. Rep. 2016). I am puzzled.
5. Page 4, line 115: 'a single drop' is not quantitative. Also 'muscone essential oil' should be '10 uM muscone'? This description is too vague.
6. Page 29, line 752: one-way ANOVA. This should be unpaired t-test? Page 32, line 794-798; This should also be unpaired t-test? Page 34, line 821-822: Fig. 6b should be unpaired t-test? Supp Fig 7: It should be ANOVA with Dunnett's test? The statistics in the entire manuscript should be re-examined.

Reviewer #2:

Remarks to the Author:

The manuscript of Wu et al describes the development and implementation of a chemically inducible gene expression system tailored-engineered for gene therapy approaches via AAV-mediated transduction in mice. In brief, the authors developed a muscone inducible switch/circuit in mammalian cells consisting of a G-protein coupled muscone activated receptor, which triggers a cAMP/PKA signal transduction cascade activating gene expression driven by a CREB1-cognate synthetic promoter. The components and associated elements were packed in a series of AAV vectors and first functionally characterised (dynamic ranges, time and chemical dose responses, etc) and further optimised in mammalian cell culture. The system was customised for the expression of a luciferase or two different therapeutic proteins for treating non-alcoholic fatty liver disease and allergic asthma, respectively. In vivo experiments in mice were performed upon targeting the transgenes to the liver in the former case or the lung in the latter. The therapeutic efficacy thereof was evaluated in well-designed experiments and a relatively complete characterization was performed.

The article is well written, clear and the graphical support adequate. The subject fits well into the profile of the journal and the research adds an alternative to the already proven useful toolbox of AAV-mediated methods to control gene expression in mammalian systems in vivo, however contributing with a very useful additional property, namely the inducibility function. The experimental strategy used by the authors is very clever and potentially useful and expandable,

either upon exchange of the inducer/receptor/specificity or the therapeutic protein and cellular targets, which is the main strength of the article, representing the key point and message of the paper. The introduction of the (functional!) tools and the proof of principle examples chosen illustrate the potential relevance for the biomedical field opening up novel perspectives for gene therapy approaches.

The main criticism to the work lies on two main aspects somehow related dealing with the molecular design and components used. These have not been properly addressed and should be more clearly discussed in the discussion. Both to instruct the reader willing to use the approaches and also to stimulate future improved systems. The first one is that the system contains several, too many components which might affect the robustness and hinder its portability and also makes it require 2-3 AAVs that have to be independently produced, transduced and of course co-transduced in the same cell(s) which also leads to reduced efficiencies. The second aspect is that the system is not orthogonal to the mammalian cells used, as it uses endogenous central signalling components and pathways and might lead to detrimental effects (cAMP, PKA, etc). The authors have analysed a reduced number of cellular/molecular proxys, however this is not comprehensive and falls short. A gene expression (RNASeq) study on the in vivo tissues is required to assess the effects on the target cells/tissues. This is necessary in virtue of the relevance of the work and the claims in terms of potential applications in gene therapy, and can be done with material already produced (therapeutic efficacy experiments).

Minor points:

- Fig 4b) the starting levels (and throughout) of deltaHGF21 for the samples treated with vehicle are high. Is this leaky expression of the system or the endogenous levels? In any event the dynamic range is just 3-fold. This review wonders what is the threshold of response in vivo.
- The intrinsic limitations of chemically inducible systems should be discussed in the article
- The characterization performed including different ratios of plasmids (Suppl Fig 1) is irrelevant for the applications with AAVs. This just informs that the performance of the system is dependent, at least in cell culture, on the relative and/or absolute amounts of the various components. For the in vivo experiments however, there is absolute no control on these parameters, which leads to the conclusion that the variability of the system in vivo is very high.
- Along these lines, are the genes integrated in the target cells upon transduction? if not, there will be a dilution upon cell replication and there would only be transient expression. There should be a report on what actually happens with the components. There is a system involving 3 and one with 2 AAVs in use. Cells will only respond when all elements are simultaneously transduced in a cell. What is the efficiency obtained, how does the distribution curve look like. This should be part of the characterization as it is a much relevant aspect of the system and to design the applications, expected outcomes, etc.
- It is not clear in experiment of figure 5 two different serotypes of virus with different targets are used? Why not to use the same? The goal is to target the same cell with both viruses.
- Sentence starting in line 232: "Collectively, these results indicate that the AAVMUSE enables long-term induction of transgene expression in the lungs." This is misleading. It should be clarified "long term induction when exposing the animal regularly to the inducer", otherwise one would think that the system is long-term activated just with one exposure... Idem in 281-283: when exposed repeatedly to muscone.
- Conclusions results lines 256-259. Are the levels produced physiologically relevant? A comparison with normal/ideal responses should be included.
- Based on the molecular design of the constructs, this reviewer wonders how the intrinsic membrane protein RTP1S is successfully targeted to the plasma membrane when it is actually expressed in a construct downstream of a 2A peptide.
- In order for the system to be applicable in gene therapy approaches the pharmacokinetics need to be compatible with the required responses. There seems not to be any data on the dynamics of expression in vivo. Only scales in the order of several days are shown. How fast is the system reacting? What are the kinetics & dynamics of expression in vivo. For an allergic reaction, fast responses would be required, it would be therefore necessary in order to evaluate the applicability of the system and support the claims to have such analyses performed/to have a dynamic description at shorter time scales (at the mRNA level would already be very useful).

Reviewer #3:

Remarks to the Author:

In the present manuscript Wu et al described the development of a muscone inducible transcription system. The initial data were generated by cell transfection and next the system was evaluated in vivo using AAV as delivery vehicles. The model is quite complex because require the delivery of 3-4 different elements however the data presented indicate that muscone responsive receptor allowed control expression in vitro and in vivo in a dose dependent manner.

Howe there is a question I would like the authors to answer: Can the system be induced by alternative drugs or endogenously produced molecules, did the author tested that?

There are additional data required for a full characterization of the system.

Figure 1 a and b should be indicated in the text.

Biodistribution analysis of transgene expression should be analyzed in the different models, AAV9 transduced very efficiently the liver but also other organs and it can even cross the blood brain barrier where Gaolf is expressed. The same for AAV6.

Luciferase, FDG21 and IL14 should be analyzed in different organs.

Luciferase expression decreases with time, is this due to the development of immune reaction?

Please indicate how long after muscone exposure DFGF21 is analyzed.

Could the authors explain why they don't observe expression when using AAV-lung and why they observe expression when using two different serotypes AAV-lung and AAV6. Does AAV6 go to the lung after iv expression. The explanation: "We speculated that the numbers or capsids of the AAV vectors affected gene delivery efficiency" is really very strange.

In the case of asthma the authors should indicate that they prevent the development of sensitivity to ova, they are not treating a chronic disease.

Reviewer #4:

Remarks to the Author:

The manuscript reports development and preliminary validation of a very interesting inducible transgene expression system.

While the system appears to be quite functional and thus has clear applications in gene therapy applications, there are number of major limitations of the system that, in opinion of this reviewer, significantly limit its translational potential. While the system can be utilised in preclinical studies, the lack of direct translability will limit the impact on the broader field of gene therapy.

In addition to the major concerns outlined below, there are also number of minor concerns / errors that need to be addressed.

Major Concerns

While the authors cite in the introductions examples where transgene expression could (did) lead to undesired toxic effects, and use those examples to justify the need to an inducible system, such toxicity has only been observed in a very small subset of gene therapy applications. Thus, while the described system is indeed quite interesting and could have very useful applications in genetic engineering, the clinical need for such system is quite unclear to the reviewer.

In addition, if such system was utilised in a clinical setting, the difficulty in finding the most optimal timing of muscone exposure (and dosing) would create a significant translational problem. This would be a major safety barrier, as 1) too much exposure could lead to transgene expression similar to that observed for a canonical gene augmentation strategies (which, as per authors' opinions, could lead to toxicity); 2) too little expression could make the therapy ineffective / suboptimal.

Finally, (and the authors pointed this as one of the current limitations) the system requires co-delivery of 2 or 3 AAV vectors. This would greatly increase the cost of therapies and would trigger safety concerns. Thus, the translational impact of the technology developed and validated in this study is limited, at least in the foreseeable future.

Minor Comments

Line 21: "...control of therapeutic outputs..." – the term "therapeutic outputs" is unclear. Please rewrite the sentences to clarify that it is referring to control of "therapeutic transgene expression", as the reviewer assumes that is what this term is referring to.

Line 26: Just few sentences later the term "control of transgene expression" is used to what I believe is the same process as in the first sentence. Just like the first term, this expression is also unclear and needs to be modified to avoid ambiguity.

Line 25: "AAVMUSE enables remote, muscone dose- and exposure-time-dependent control of transgene expression in livers or lungs of mice for at least 20 weeks, as activated by muscone, a macrocyclic compound of musk." – no need to state twice that the control is related to "muscone dose" and "activated by muscone" as to this reviewer it is redundant.

Line 31: "...can support future gene therapies..." – this is an overstatement. The odorant-molecule-controlled system has not supported any gene therapies to date, so it cannot support "further" therapies...

Line 42: "...adeno-associated viruses (AAVs) are gaining popularity as a platform for gene delivery to treat a variety of human diseases,...". Viruses are not used to deliver genes (transgenes) to treat diseases. We use "vectors" for that. Please rewrite the sentence to state: "Owing to low immunogenicity and lack of pathogenicity, recombinant vectors based on adeno-associated viruses (rAAVs) are gaining popularity..."

Line 46: the term "recombinant proteins" is used twice in the same sentence. Please rewrite.

Line 66: Please replace "controllable gene therapy" with "controllable expression of gene therapeutic", to clarify that it is the level of transgene expression that is being controlled.

Figures should be cited in order of their appearance. While Figure 1 is called in Line 74, the next reference is to Figure 1c (Line 88). Figure 1a and 1b are not referred to.

Line 151: The authors state that the effects were observed for up to 28 weeks in liver or even 31 weeks (line 195). In the lung the results are reported up to week 20. In the abstract the authors state that "...control of transgene expression in livers or lungs of mice for at least 20 weeks", while in the discussion (line 283) it is stated that "...can induce reporter gene expression after a single AAV administration over 20 weeks". While none of those statements is actually wrong, it would be probably beneficial if the authors could be more consistent with their conclusion(s).

Line 209: "We speculated that the numbers or capsids of the AAV vectors affected gene delivery efficiency." What does the term "numbers" refer to? Why is "capsids" plural, since to the reviewer's understanding the same capsid was used for all three constructs.

Line 214: "We eventually selected concatenated construct pWX325 (ITR-PSV40-RTP1S::pA-SEAP-PCRE-pA-ITR)." – can the authors provide more information as to what "eventually" led to the selection of pWX325 over pWX322.

Line 215-217: please make it clear that this refers to a transient transfection of plasmid constructs and now AAV transduction. It may be useful to clarify statement in line 211: "Seeking to facilitate in vivo delivery, we first simplified AAVMUSE into two separate AAV vector plasmids..."

Lines 219-225: it is unclear to the reviewer why the authors had to use two AAV variants (AAV2/lung and AAV2/6). Both constructs (pWX126 and pWX342). Have to transduce the same cell for this system to work. Why was that not possible using the same AAV variant for both constructs (either AAV2/lung or AAV2/6)?

Manuscript number: NCOMMS-23-16959-T—Point-by-point responses to referees' comments:

We want to thank the editor and the referees for their constructive comments. As shown below, we have undertaken a careful revision process to substantially improve our study's scientific rigor and impact. We presented point-by-point responses to each of the referee comments (below). We started with a bulleted summary of the major experiments performed during our revision process.

- 1) performed a gas chromatography-mass spectrometry (GC-MS) analysis to determine the concentration of muscone in the liver, lung, and plasma from mice exposed to nebulized muscone, as well as in the headspace of ultrasonic nebulizer (**new Fig. S10**).
- 2) performed an *in vitro* experiment to investigate whether the olfactory receptor MOR215-1 of the MUSE transgene system is specific to muscone. Our new data showed that MOR215-1 could be activated only by muscone structural analogs and was completely insensitive to the endogenously produced molecules that could cross-activate the cAMP signals (**new Fig. S2 and S3**).
- 3) conducted an RNA-seq study to assess the effects of the AAV_{MUSE} system on the target tissues at the molecular level. Using Gene Set Enrichment Analysis (GSEA), our new data showed that cAMP pathway-related genes were not differential enrichment in AAV_{MUSE}-transduced NAFLD mice livers compared to control mice (**new Fig. S15**).
- 4) performed animal experiments to indicate that the combination of AAV2/6 and AAV2/lung resulted in higher bioluminescence signals in lungs compared to a single serotype alone (**new Fig. S21**).
- 5) performed a luciferase expression experiment and qPCR analysis to assess the dynamics of the MUSE system *in vitro*. Our new data showed that the MUSE system exhibited fast kinetics, inducing a significant luciferase signal within 2 hours (**new Fig. S7**).
- 6) performed new animal experiments to characterize AAV_{MUSE}-mediated luciferase expression kinetics. Our new data showed that 2-hour nebulized muscone exposure is sufficient to trigger luciferase expression, with the maximum luciferase expression at 4 hours (**new Fig. S11**).
- 7) performed new experiments to investigate the biodistribution of transgene expression in AAV_{MUSE}-transduced mice. We observed a significantly higher luciferase signal in the liver than in other organs (**new Fig. S9**).

Response to Reviewers' Comments:

Reviewer #1 (Remarks to the Author):

Taking advantage of a volatile non-invasive character of an odorant, the authors developed a system to express a specific olfactory receptor in a target tissue in the mouse body using adeno-associated virus vector and stimulate the receptor by letting the infected mouse inhale the ligand odorant. This system successfully allowed for the induction of a long-term controllable expression of a desired proteins associated with two chronic inflammatory diseases. This is a unique gene therapy system that could be widely used for a specifically targeted precision therapy for various human diseases.

Thank you very much for your highly positive comments.

1. Since mice express endogenously MOR215-1 in the nose, it should be confirmed that various therapeutic effects are via an AAV-delivered MOR215-1 expressed in liver or lung tissues but not via an olfactory tissue-expressed MOR215-1. The easiest way is to examine the effects in MOR215-1 null mice.

Reply:

Outstanding points. We agree with the reviewer that ruling out the contribution of endogenous MOR215-1 is important. Instead of using MOR215-1 null mice, we used an alternative strategy to address this issue. Specifically, NAFLD model mice, transduced with the control reporter luciferase (AAV_{Luc}), were exposed to the same muscone regime as the AAV_{MUSE-ΔhFGF21} group (**Figure 4a**). Since AAV_{Luc}-transduced NAFLD model mice, with or without exposure to muscone, are comparable to AAV_{MUSE-ΔhFGF21}-transduced NAFLD model mice exposed to vehicle (**Figure 4 and Supplementary Fig. 17**), we concluded that endogenous MOR215-1 in the nose has no significant effects.

2. The muscone concentration in the headspace of an ultrasonic nebulizer should be measured. Another question in this line is how much muscone is delivered to the liver and lung. You can extract the tissue with an organic solvent and measure the amount of muscone by GC-MS, or can measure the amount of muscone in the blood. The amount should be consistent with the EC50 value for MOR215-1.

Reply:

Thanks for this comment. As suggested, we have measured the muscone concentration by GC-MS in different regions of the mice, including mouse liver, lung, plasma, and the headspace of an ultrasonic nebulizer. The new results showed that the muscone concentration was 332 nM, 444 nM, 722 nM, and 0.18 mM in mouse livers, lungs, plasmas, and the headspace of an ultrasonic nebulize, respectively, which is within the range of the EC50 of MOR215-1, reported by Kazushige Touhara group (**new Supplementary Fig. 10**).

We have added this information in the revised Supplementary Information document. Please see page 11 and below:

Supplementary Fig. 10 Quantification of the muscone concentration by GC-MS. (a) Muscone quantification in the livers, lungs, and plasma of mice exposed to nebulized muscone using an ultrasonic nebulizer. (b) Measurement of muscone in the headspace of ultrasonic nebulizer by GC-MS. Data are presented as means \pm SEM ($n = 3$ mice). P values were obtained from two-tailed unpaired t -tests. $*P < 0.05$, $**P < 0.01$.

We have added the associated methods in the revised “Methods” part. Please see pages 18 and 19, and below:

Methods

I. Instruments and mass spectrometric conditions

The quantitative analysis was performed on an Agilent 7890B gas chromatography system coupled with an Agilent 7000D triple quadrupole mass spectrometer (Agilent Technologies, USA). The processed samples were placed in the vials of the autosampler for injection. The chromatographic column was the DB-5UI capillary column (30 m \times 0.25 mm, 0.25 μ m). The separation procedure of the GC system was set at the initial temperature of 80°C held for 1 minute, then heated up to 300°C at a speed of 10°C/minute and held for 5 minutes. No splitting mode was set. Helium was the carrier gas with a 1.0 mL/minute flow rate. The temperature of interface, ion source, and Quadrupole temperature were set at 280°C, 230°C, and 150°C, respectively.

II. Preparation of standard and sample solutions

The stock solution of muscone (20 μ g/mL) was prepared in methanol and diluted with ethyl acetate to produce 0-1000 ng/mL working standard solution used in the linearity experiment. The internal standard (IS) working solution (30 μ g/mL) was prepared by diluting Naphthalene stock solutions (2 mg/mL) with ethyl acetate. The Calibration standards were prepared by spiking standard working solution (10 μ L) and IS working solution (10 μ L) with blank plasma (90 μ L) to obtain a serial of muscone concentrations (0, 1, 3, 5, 10, 30, and 100 ng/mL). The sample treatment was a simple liquid-liquid extraction (LLE). The 0.1 mL mouse blood was collected from the orbital venous plexus and transferred to 1.5-mL EDTA-polythene tubes 4 hours after exposure to nebulized muscone. The plasma samples were prepared by spiking 50 μ L plasma, 10 μ L IS, 50 μ L blank plasma with 200 μ L neat ethyl acetate. The livers and lungs were dissected from mice 4 hours after exposure to nebulized muscone. The liver and lung samples were

prepared by spiking 100 μL sample, 10 μL IS with 200 μL neat ethyl acetate. The concentration of muscone in the headspace of the ultrasonic nebulizer was analyzed after diluting by 1000 times, and then 100 μL diluted muscone was added with 10 μL IS and 200 μL ethyl acetate. Then, the mixture solution was vortexed for 1.0 min and centrifuged at $13000 \times g$ at 4°C for 10 min. The supernatant was transferred into an injection vial for analysis. All the prepared solutions were stored at -20°C until analysis. 1.0 μL supernatant was subjected to GC-MS/MS system for analysis.

3. To show that the induction of AAV-mediated transgene expression is indeed mediated by MOR215-1, non-ligand for MOR215-1, for example, described in Sato-Akuhara et al. *J. Neurosci.* 2016, should be tested as a negative control. Having results in this experiment, one can also suggest that the use of an olfactory receptor is advantageous because of the high ligand specificity.

Reply:

Thanks for this helpful suggestion. As suggested, we have performed additional experiments to address the specificity of the olfactory receptor MOR215-1 of MUSE transgene system. In this study, the non-ligands for MOR215-1 (galaxolide, tonalide, celestolide, cyclopentadecane, and 8-pentadecanone), reported by Narumi Sato-Akuhara (Sato-Akuhara et al., *J. Neurosci.*, 2016), were selected as negative controls. The new data showed that two structural analogs (including cyclopentadecanone and cyclopentadecanol) triggered SEAP production, while the non-ligands did not (cross)-trigger its production in MOR215-1/CRE transgenic HEK-293T cells (**new Supplementary Fig. 2**). These results further support the view that the system is highly specific to a narrow and readily controllable substrate window and the risk of unspecific activation is low.

We have added this information in the revised Supplementary Information document. Please see **page 3, and below**:

Supplementary Fig. 2 Cross-reactivity of AAV_{MUSE} to musk-related chemicals. pWX126/pWX127/pWX123-transgenic HEK-293T cells were incubated with muscone [C₁₆H₃₀O], cyclopentadecanone [C₁₅H₂₈O], cyclopentadecanol [C₁₅H₃₀O], galaxolide [C₁₈H₂₆O], tonalide [C₁₈H₂₆O], celestolide [C₁₇H₂₄O], cyclopentadecane [C₁₅H₃₀] and 8-pentadecanone [C₁₅H₃₀O]) at 10 μM. SEAP expression was profiled in the cell culture supernatant after 24 hours. Data are presented as means ± SD; *n* = 3 biologically independent samples. All plasmids are described in Supplementary Tables S1 and S3.

We have added the associated methods in the revised “Methods” part. Please see **page 15 and below: Chemicals**

Odorants utilized in this study were purchased from Macklin Biochemical Technology Corporation. Castor oil was purchased from Aladdin. The musk odorants and musk analogs were prepared as 200 mM stock solutions in dimethyl sulfoxide or anhydrous ethanol. Menthol (catalog no. HY-75161), dopamine (catalog no. HY-B0451A), and CU-T12-9 (catalog no. HY-110353) were purchased from MedChemExpress (China) and freshly dissolved in water before each experiment. GLP-1 (catalog no. G3265), atrial natriuretic peptide (ANP) (catalog no. A1663), and human insulin (catalog no. I9278) were purchased from Sigma-Aldrich (Buchs, Switzerland).

4. In the lung experiments, the response was seen without Golf, which is in contrast with the case of the liver. Does this mean that the MOR215-1 is coupling with endogenous Gs? Then, why is Golf necessary in liver that should also express Gs? It has been known that the lung expresses Golf (Aisenberg, Sci. Rep. 2016). I am puzzled.

Reply:

Thanks for pointing this out, we apologize for the confusion caused. We clarified the confusion as follows: our data from cell culture experiments showed that the combination of four plasmids encoding the muscone-responsive vector pMOR215-1, the cAMP-responsive promoter (P_{CRE}) driven reporter gene vector pCK53, the truncated version of the receptor-transporting protein vector pRTP1S, and the olfactory neuron-specific G protein alpha subunit vector pG_{α_{olf}} induced SEAP expression by 28-fold, while the combination of **three plasmids** encoding pMOR215-1, pCK53, and pRTP1S induced SEAP expression to a similar extent (24-fold) (**Supplementary Fig. 1a**). This data suggested that MUSE system remains inducible by muscone even in the absence of G_{α_{olf}} component, which might suggest that the MOR215-1 was coupling with endogenous Gs.

To deliver the MUSE system for *in vivo* application, we chose a relatively simple system comprising a few constructs for mouse injection. We consolidated the constructs of G_{α_{olf}} and RTP1S into a single vector (pWX127). Then mice were transduced with AAV2/9 vectors carrying the AAV_{MUSE} system components [the concatenated vector pWX127 (ITR-P_{SV40}-G_{α_{olf}}-P2A-RTP1S-pA-ITR), pWX126 (ITR-P_{SV40}-MOR215-1-pA-ITR), and the reporter vector pWX158 (ITR-P_{CRE}-luciferase-P2A-EGFP-pA-ITR)] via tail vein injection. Two weeks after the AAV injection, AAV_{MUSE}-transduced mice

exposed to nebulized muscone showed high bioluminescence signal intensities in the liver, compared to vehicle-exposed control mice transduced with AAV_{MUSE} (**Supplementary Fig. 8**). These results demonstrate that AAV_{MUSE} can efficiently induce luciferase expression in mouse livers.

To test whether AAV_{MUSE} induced transgene expression in mouse lungs, we initially used AAV2/lung to package AAV_{MUSE} containing three AAV vectors: pWX126 (ITR-P_{SV40}-MOR215-1-pA-ITR), pWX127 (ITR-P_{SV40}-G α_{olf} -P2A-RTP1S-pA-ITR), and pWX158 (ITR-P_{CRE}-luciferase-P2A-EGFP-pA-ITR). However, no luciferase signal was detected in the lung (**Supplementary Fig. 19**). We speculated the number of AAV_{MUSE} component vectors or capsid of the AAV vectors reduced gene delivery efficiency.

So, we next simplified AAV_{MUSE} into two separate AAV vectors: a muscone-responsive vector pWX126 and a concatenated vector for two expression cassettes (pWX322, ITR-P_{SV40}-RTP1S-pA::P_{CRE}-SEAP-pA-ITR or pWX325, ITR-P_{SV40}-RTP1S::pA-SEAP-P_{CRE}-ITR) (**Supplementary Fig. 20a, b**). We eventually selected construct pWX325 (ITR-P_{SV40}-RTP1S::pA-SEAP-P_{CRE}-ITR), which enabled induction of SEAP expression independent of the G α_{olf} component. Further, considering that a combination of AAV serotypes has been shown to enhance gene transduction (Aubert et al., *Nat. Commun.*, 2020), we performed additional experiments to demonstrate that the combination of two AAV serotypes (AAV2/lung and AAV2/6) indeed improved gene transduction to lungs (**new Supplementary Fig. 21**).

Next, we combined two AAV serotypes to deliver AAV_{MUSE}: pWX126 packaged into an AAV2/lung capsid (which was reported to efficiently transduce the endothelium of the pulmonary vasculature) and pWX342 packaged into an AAV2/6 capsid (which was reported to efficiently transduce both airway and alveolar type II cells). We found that AAV_{MUSE}-transduced mice exposed to nebulized muscone exhibited significantly higher bioluminescence signal intensities (up to 17-fold induction) in lungs compared to vehicle-exposed control mice transduced with AAV_{MUSE} (**Fig. 5c, d**). These results indicate that the AAV_{MUSE} without the G α_{olf} component enables the induction of luciferase expression in mouse lungs.

5. Page 4, line 115: ‘a single drop’ is not quantitative. Also, ‘muscone essential oil’ should be ‘10 μ M muscone’? This description is too vague.

Reply:

Thanks for this helpful suggestion. We have corrected to “10 μ M muscone” in the revised manuscript. Please see **page 5, line 129**.

6. Page 29, line 752: one-way ANOVA. This should be unpaired t-test? Page 32, lines 794-798; This

should also be unpaired t-test? Page 34, line 821-822: Fig. 6b should be unpaired t-test? Supp Fig 7: It should be ANOVA with Dunnett's test? The statistics in the entire manuscript should be re-examined.

Reply:

Thank you so much for the suggestion. We have re-examined all the statistical analysis in the revised manuscript. Please see the legends of Figures 3, 4, 5, and 6 and new Supplementary Figures 9, 14, 16-18, and 21.

Reviewer #2 (Remarks to the Author):

The manuscript of Wu et al describes the development and implementation of a chemically inducible gene expression system tailored-engineered for gene therapy approaches via AAV-mediated transduction in mice. In brief, the authors developed a muscone inducible switch/circuit in mammalian cells consisting of a G-protein coupled muscone activated receptor, which triggers a cAMP/PKA signal transduction cascade activating gene expression driven by a CREB1-cognate synthetic promoter. The components and associated elements were packed in a series of AAV vectors and first functionally characterised (dynamic ranges, time and chemical dose responses, etc) and further optimised in mammalian cell culture. The system was customised for the expression of a luciferase or two different therapeutic proteins for treating non-alcoholic fatty liver disease and allergic asthma, respectively. *In vivo* experiments in mice were performed upon targeting the transgenes to the liver in the former case or the lung in the latter. The therapeutic efficacy thereof was evaluated in well-designed experiments and a relatively complete characterization was performed.

Reply:

Thank you very much for your highly positive comments.

The article is well written, clear and the graphical support adequate. The subject fits well into the profile of the journal and the research adds an alternative to the already proven useful toolbox of AAV-mediated methods to control gene expression in mammalian systems *in vivo*, however contributing with a very useful additional property, namely the inducibility function. The experimental strategy used by the authors is very clever and potentially useful and expandable, either upon exchange of the inducer/receptor/specificity or the therapeutic protein and cellular targets, which is the main strength of the article, representing the key point and message of the paper. The introduction of the (functional!) tools and the proof of principle examples chosen illustrate the potential relevance for the biomedical

field opening up novel perspectives for gene therapy approaches.

Reply:

Thank you very much for your highly positive comments.

The main criticism to the work lies on two main aspects somehow related dealing with the molecular design and components used. These have not been properly addressed and should be more clearly discussed in the discussion. Both to instruct the reader willing to use the approaches and also to stimulate future improved systems. The first one is that the system contains several, too many components which might affect the robustness and hinder its portability and also makes it require 2-3 AAVs that have to be independently produced, transduced and of course co-transduced in the same cell(s) which also leads to reduced efficiencies.

Reply:

Thank you for your highly constructive suggestions to help clarify our study.

Due to the limitation of packaging size of AAV vectors (4.7k bp), we have to design our MUSE system with three or two separate viral vectors for gene therapy in diet-induced NAFLD model mice or allergic asthma model mice. Despite using three or two AAVs, our data showed that AAV_{MUSE-ΔhFGF21} induced therapeutic levels of ΔhFGF21 to treat NAFLD in mice (**Fig. 4**) and AAV_{MUSE-ΔmIL-4} can control ΔmIL-4 to inhibit eosinophil infiltration and pro-inflammatory cytokines and to prevent the development of allergic asthma in mice with one single injection (**Fig. 6**).

We totally agree with the reviewer that multiple components in our system obviously hinder its wide application in future translational studies. An ideal solution is to develop a more packable device with a small construct size to ensure efficient delivery *in vivo*. In the future, artificial intelligence (AI)-guided protein structure prediction and classification will be used to predict the structures of MUSE modules: RTP1S and MOR215-1 and generate the miniature variants with good responsive capability to muscone (Huang et al., *Cell*, 2023). In addition, a smaller version of MUSE suitable for packaging within a single AAV will be engineered by truncating and mutating the MUSE modules via high-throughput ligand screening (Xiang et al., *Nat. Biotechnol.*, 2023 and Yasi et al., *Curr. Opin. Biotechnol.*, 2020). Another idea for the efficient *in vivo* delivery of MUSE is using engineered virus-like particle (eVLP) developed by David R Liu (Banskota et al., *Cell*, 2022), which holds great promise as vehicles for the targeted delivery of therapeutic macromolecules and represents an improvement over AAV or plasmid delivery.

We have included this information about the potential for AAV_{MUSE} system improvement and optimization in the revised “Discussion” part. Please see **page 13** and **below**:

“Moreover, multiple components of our system are an obstacle for wide application. Possible solutions include designing a smaller version of MUSE suitable for packaging within a single AAV by truncating and mutating the AAV_{MUSE} modules containing RTP1S and MOR215-1, perhaps based on artificial intelligence (AI)-guided protein structure prediction and classification or via high-throughput ligand screening. Another idea for the efficient *in vivo* delivery of MUSE is using engineered virus-like particles (eVLP) (Banskota et al., *Cell*, 2022), which hold great promise as vehicles for targeted therapeutic macromolecules delivery and represent an improvement over AAV or plasmid delivery.”

The second aspect is that the system is not orthogonal to the mammalian cells used, as it uses endogenous central signaling components and pathways and might lead to detrimental effects (cAMP, PKA, etc). The authors have analysed a reduced number of cellular/molecular proxys, however this is not comprehensive and falls short.

A gene expression (RNASeq) study on the *in vivo* tissues is required to assess the effects on the target cells/tissues. This is necessary in virtue of the relevance of the work and the claims in terms of potential applications in gene therapy, and can be done with material already produced (therapeutic efficacy experiments).

Reply:

Thanks for this comment. We found no apparent crosstalk between this system and major signaling pathways, including mitogen-activated protein kinase (MAPK)-mediated protein kinase (IRS-1-Ras-MAPK) pathway through the activation of insulin receptor (IR) (Ye et al., *Nat. Biomed. Eng.*, 2017), intracellular calcium-dependent nuclear factor of activated T cells (NFAT) pathway through the activation of transient receptor potential (TRP) melastatin 8 (TRPM8) (Bai et al., *Nat. Med.*, 2019), and proinflammatory nuclear factor NF-κB activation in response to toll-like receptor 2 (TLR2) signaling (Li et al., *Int. Immunopharmacol.*, 2023) (**new Supplementary Fig. 5**).

As suggested, we have conducted an RNASeq study to assess the effects on the target tissues (livers) of control mice and mice exposure to muscone. GSEA (Gene Set Enrichment Analysis) (Wang et al., *Immunity*, 2023) showed that cAMP pathway-related genes from the KEGG (Kyoto Encyclopedia of Genes and Genomes) were not differentially enriched in the livers of AAV_{MUSE}-transduced mice compared to control mice (AAV_{MUSE}-transduced mice exposed to vehicle or un-transduced wild-type mice), which indicated that our system did not cause detrimental effects on the endogenous target genes of the cAMP signaling pathway (**new Supplementary Fig. 15**).

These results indicate that our system is robust and orthogonal. Our system can adapt to various host physiological conditions without affecting endogenous molecules or signaling pathways. This feature is highly desirable for long-term gene regulation applications using the MUSE system in gene therapy.

We have included this description in the revised Supplementary Information document. Please see pages 6 and 18, and below:

Supplementary Fig. 5 Interference from major cellular signaling pathways with AAV_{MUSE}. pWX126/pWX127/pWX123-transgenic HEK-293T cells were transfected with the plasmids encoding IR, TRPM8, or TLR2, and incubated with their respective agonists (insulin 20 ng/mL; menthol, 50 μ M; CU-T12-9, 1 μ M) in the presence or absence of muscone. SEAP expression was profiled after 24 hours. Data are presented as means \pm SD; $n = 3$ biologically independent samples. P values were obtained from two-tailed unpaired t -tests. n.s., not significant. All plasmids are described in Supplementary Tables S1 and S3.

Supplementary Fig. 15 Gene Set Enrichment Analysis (GSEA) for cAMP pathway in the different mouse groups. (a) GSEA for cAMP pathway in the livers of AAV_{MUSE}-transduced mice exposed to nebulized muscone versus control wild-type mice. (b) GSEA for cAMP pathway in the livers of AAV_{MUSE}-transduced mice exposed to

nebulized muscone versus AAV_{MUSE}-transduced mice exposed to vehicle. FDR: false discovery rate. $n = 5$ mice.

We have added the associated methods in the revised “Methods” part. Please see **page 17**, and **below: RNA-sequencing**

Livers were dissected from AAV_{MUSE}-transduced mice exposed to muscone and control mice (AAV_{MUSE}-transduced mice exposed to vehicle or un-transduced wild-type mice) and resuspended in TRIzol reagent (catalog no. 15596026, Invitrogen). Total RNA was extracted and treated with DNase I (Qiagen, Dusseldorf). Library preparation, quality control, and sequencing were performed by Genechem (Shanghai, China). RNA sequencing was performed via the Illumina platform (Illumina Novaseq 6000, CA) based on the mechanism of SBS (sequencing by synthesis). Gene set enrichment analysis (GSEA) was performed on the local tool <http://www.broadinstitute.org/gsea/index>. The raw data of RNA-seq was uploaded into the NCBI Sequencing Read Archive (SRA) under accession number PRJNA1011482.

Minor points:

1. - Fig 4b) the starting levels (and throughout) of Δh FGF21 for the samples treated with the vehicle are high. Is this leaky expression of the system or the endogenous levels? In any event the dynamic range is just 3-fold.

This review wonders what is the threshold of response *in vivo*.

Reply:

Thank you for bringing up this point. In the initial stage of this study, we measured Δh FGF21 levels in plasma of NAFLD model mice transduced with the control AAV_{Luc} (2×10^{11} vg) via tail vein and found the concentration of Δh FGF21 was around 5 ng/mL, consistent with a previous report by Veronica Jimenez (Davidsohn et al., *Proc. Natl. Acad. Sci. U S A.*, 2019 and Jimenez et al., *EMBO Mol. Med.*, 2018).

However, the concentration of Δh FGF21 was around 10 ng/mL in NAFLD model mice transduced with AAV_{MUSE- Δh FGF21} and exposed to vehicle. We think that the AAV_{MUSE} system exhibits a certain degree of leakiness. However, such a low level of Δh FGF21 expression has no impact on the body weight of NAFLD mice transduced with AAV_{MUSE- Δh FGF21} in the absence of muscone. The therapeutic threshold of FGF21 *in vivo* was above 20 ng/mL, which was also reported by Fatima Bosch (Jimenez et al., *EMBO Mol. Med.*, 2018). In our study, the AAV_{MUSE}-induced Δh FGF21 expression level reached a range of 20-30 ng/mL, which is within the desired therapeutic window.

2. - The intrinsic limitations of chemically inducible systems should be discussed in the article

Reply:

Thanks for the constructive comment. We aim to comprehensively discuss the advantages and challenges of our chemically-inducible MUSE system. We discussed the intrinsic limitations of

chemically inducible systems in the revised “Discussion” section. Please see **page 13**, and **below**:

“Despite our successful demonstrations of treating chronic diseases using AAV_{MUSE}-based gene therapies, some barriers must be overcome before chemically inducible systems can be widely accepted for a broad range of diseases, such as the potential cytotoxicity from the chemical inducers that affect cell numbers and colony formation (Ermak et al., *Anal. Biochem.*, 2003), and the stability and metabolism of the chemical inducer that affects the duration and consistency of gene expression or diffusing freely *in vivo*.”

3. -The characterization performed, including different ratios of plasmids (Suppl Fig. 1), is irrelevant for the applications with AAVs. This just informs that the performance of the system is dependent, at least in cell culture, on the relative and/or absolute amounts of the various components. For the *in vivo* experiments however, there is absolute no control on these parameters, which leads to the conclusion that the variability of the system *in vivo* is very high.

Reply:

Thanks for your comments. Our data from cell culture experiments using different ratios of AAV_{MUSE} system components showed that the higher amount of receptor MOR215-1 resulted in higher induction (63-fold) of SEAP expression in the presence of muscone in the culture medium (**Supplementary Fig. 1b**).

Thus, in the initial stage of our study, we performed experiments in three groups of mice using different ratios of AAV titers, including Group 1: the muscone-responsive vector AAV2/9-pWX126 (2×10^{11} vg), AAV2/9-pWX127 (2×10^{11} vg), AAV2/9-pWX158 (2×10^{11} vg); Group 2: AAV2/9-pWX126 (3×10^{11} vg), AAV2/9-pWX127 (1.5×10^{11} vg), AAV2/9-pWX158 (1.5×10^{11} vg); and Group 3: AAV2/9-pWX126 (3.6×10^{11} vg), AAV2/9-pWX127 (1.2×10^{11} vg), AAV2/9-pWX158 (1.2×10^{11} vg). 2 weeks after AAV injection, AAV_{MUSE}-transduced mice were exposed to nebulized muscone (18 mM) for 4 hours and then bioluminescence imaging was quantified using an *in vivo* imaging system. The results showed that there was lower luciferase expression in Group 1 (1:1:1 ratio) compared to Group 2 (2:1:1 ratio) and Group 3 (3:1:1 ratio). Moreover, there was no difference in luciferase expression between Group 2 and Group 3. Therefore, we selected the 2:1:1 ratio for further animal experiments.

AAV_{MUSE}-mediated luciferase expression in mouse livers. (a) *In vivo* bioluminescence images of the BALB/c mice transduced with different component ratios of AAV_{MUSE} packaged into AAV2/9 capsid. (b) Bioluminescence measurements of the muscone-induced luciferase expression based on bioluminescence imaging in a. Data in b were presented as means ± SEM ($n = 3$ mice). P values were obtained from two-tailed unpaired t -tests. $*P < 0.05$. n.s., not significant.

4. - Along these lines, are the genes integrated into the target cells upon transduction? if not, there will be a dilution upon cell replication and there would only be transient expression. There should be a report on what actually happens with the components. There is a system involving 3 and one with 2 AAVs in use. Cells will only respond when all elements are simultaneously transduced in a cell. What is the efficiency obtained, how does the distribution curve look like. This should be part of the characterization as it is a much relevant aspect of the system and to design the applications, expected outcomes, etc.

Reply:

Thanks for this comment. Adeno-associated virus (AAVs) is a promising gene therapy vector in biological research and in clinical settings, owing to their low immunogenic potential and their reduced oncogenic risk from host-genome integration (Schultz et al., *Mol. Ther.*, 2008). Despite the promise of this approach, there is a shared challenge: the transduced vector genomes are gradually lost in mitotic cells due to their persisting as non-replicating circularized episomes (Wang et al., *Nat. Rev. Drug Discov.*, 2019).

However, our MUSE system can maintain the robust luciferase expression in livers for up to 28 weeks, when exposed repeatedly to muscone (Fig. 3e, f). Moreover, the AAV_{MUSE}-transduced mice exposed

to nebulized muscone have been shown to express the luciferase reporter in lungs for up to 20 weeks (Fig. 5c, d). Collectively, these results indicate that the integration of the AAV_{MUSE} system utilizing 3 or 2 AAVs is highly efficient in the liver or lung, and this system is well preserved for a long period without a visible dilution effect in mice, which helped address the reviewer's concerns related to gene integration and efficiency in target cells.

To enhance the transduction efficiency of the AAV_{MUSE} system *in vivo*, an ideal solution is to develop a smaller version of MUSE suitable for packaging within a single AAV. In the future, AI-guided protein structure prediction and directed evolution techniques can generate miniature variants of AAV_{MUSE} modules with higher efficiency responsive to muscone.

We also have included this information in the revised “Discussion” part. Please see **page 13 and below**:

“Moreover, multiple components of our system are an obstacle for wide application. Possible solutions include designing a smaller version of MUSE suitable for packaging within a single AAV by truncating and mutating the AAV_{MUSE} modules containing RTP1S and MOR215-1, perhaps based on artificial intelligence (AI)-guided protein structure prediction and classification or via high-throughput ligand screening. Another solution for the efficient *in vivo* delivery of MUSE is using engineered virus-like particles (eVLP) (Banskota et al., *Cell*, 2022), which hold great promise as vehicles for targeted therapeutic macromolecules delivery and represent an improvement over AAV or plasmid delivery.”

5. - It is not clear in experiment of figure 5 two different serotypes of virus with different targets are used? Why not to use the same? The goal is to target the same cell with both viruses.

Reply:

Thanks for your comments. It has been demonstrated previously that using a combination of AAV serotypes could facilitate efficient gene delivery for better gene expression (Aubert et al., *Nat. Commun.*, 2020). Initially, adult BALB/c mice were transduced with luciferase reporter using three AAV serotypes (AAV2/6 alone, AAV2/lung alone, and a combination of AAV2/6 and AAV2/lung). Two weeks after the AAV injection, the bioluminescence signal was quantified using an *in vivo* imaging system. Consistent with previous findings, our results showed that the combination of AAV2/6 and AAV2/lung resulted in higher luciferase expression in mouse lungs than AAV2/6 alone or an AAV2/lung alone (**new Supplementary Fig. 21**).

Consequently, we combined two AAV serotypes to deliver AAV_{MUSE}: pWX126 packaged into an AAV2/lung capsid and pWX342 packaged into a AAV2/6 capsid. Two weeks after AAV injection, AAV_{MUSE}-transduced mice were exposed to nebulized muscone for 4 hours every six weeks or exposed to a vehicle (a mixture of castor oil and ddH₂O) (Fig. 5a, b). We found that AAV_{MUSE}-transduced mice

exposed to nebulized muscone exhibited significantly higher bioluminescence signal intensities (up to 17-fold induction) in lungs compared to vehicle-exposed control mice transduced with AAV_{MUSE} (Fig. 5c, d). Future work will focus on identifying the optimal combinations of AAV in other model systems, which would facilitate future clinical applications.

We also have included this information in the revised Supplementary Information document. Please see page 27, and below:

Supplementary Fig. 21 Assessing the impact of AAV serotype combinations on luciferase expression in mice. The BALB/c mice were transduced with luciferase reporter using AAV2/6 (2×10^{11} vg), or AAV2/Lung (2×10^{11} vg), or dual AAV serotype combination (AAV2/6, 1×10^{11} vg and AAV2/lung, 1×10^{11} vg) by tail vein injection. (a) *In vivo* bioluminescence images of the BALB/c mice transduced with three different AAV vectors packaging luciferase (ITR- P_{CMV} -luciferase-pA-ITR) at 2 weeks after the AAV injection. (b) Bioluminescence measurements of the luciferase expression based on bioluminescence imaging in a. Data in b are presented as means \pm SEM ($n = 3$ mice). P values were obtained from two-tailed unpaired t -tests. **** $P < 0.0001$. Source data are provided as a Source Data file.

6. -Sentence starting in line 232: “Collectively, these results indicate that the AAVMUSE enables long-term induction of transgene expression in the lungs.” This is misleading. It should be clarified “long term induction when exposing the animal regularly to the inducer”, otherwise one would think that the system is long-term activated just with one exposure... Idem in 281-283: when exposed repeatedly to muscone.

Reply:

Thanks for this comment. We have revised as suggested. Please see pages 6, 10, and 12, and lines 172, 272 and 324.

7. - Conclusions results lines 256-259. Are the levels produced physiologically relevant? A comparison with normal/ideal responses should be included.

Reply:

Thank you for this suggestion. We quantified IL-4 levels in the plasma of normal mice using a mouse IL-4 ELISA kit and found that IL-4 was undetectable.

Quantification of the levels of Δ mIL-4 in plasma. BALB/c mice were transduced with AAV_{MUSE} containing AAV2/lung-pWX126 (ITR-P_{SV40}-MOR215-1-pA-ITR, 5×10^{11} vg) and AAV2/6-pWX345 (ITR-P_{SV40}-RTP1S::pA- Δ mIL-4-P_{CRE}-ITR, 5×10^{11} vg) via tail vein injection. On day 14, the AAV_{MUSE}-transduced mice were exposed to a nebulized muscone or vehicle (a mixture of castor oil and ddH₂O) for 4 hours. Then, the plasma of wild-type mice and AAV_{MUSE}-transduced mice were collected, and the levels of Δ mIL-4 in plasma were quantified using a mouse IL-4 ELISA kit.

8. - Based on the molecular design of the constructs, this reviewer wonders how the intrinsic membrane protein RTP1S is successfully targeted to the plasma membrane when it is actually expressed in a construct downstream of a 2A peptide.

Reply:

Thanks for this comment. It is clear that the low induction (3-fold) of SEAP expression when HEK293-T cells are not transfected with the pRTP1S plasmid, even in the presence of G α_{olf} (**Supplementary Fig. 1a**). However, when RTP1S was positioned downstream of P2A, luciferase expression was induced by around 22-fold to 67-fold, indicating that a 2A peptide did not affect its targeting to the plasma membrane (**Supplementary Fig. 6**).

9. - In order for the system to be applicable in gene therapy approaches the pharmacokinetics need to be compatible with the required responses. There seems not to be any data on the expression dynamics *in vivo*. Only scales in the order of several days are shown. How fast is the system reacting? What are the kinetics & dynamics of expression *in vivo*. For an allergic reaction, fast responses would be required, it would be therefore necessary to evaluate the applicability of the system and support the claims to have such analyses performed/to have a dynamic description at shorter time scales (at the mRNA level would already be very useful).

Reply:

Thanks for this comment. We have performed new experiments to investigate the kinetics & dynamics of expression at shorter time scales.

First, we assessed the system's dynamics and muscone response rate in cells. HEK-293T cells were co-transfected with AAV_{MUSE}-encoding plasmids and exposed to 10 μ M muscone for various time periods (0-24 hours). This AAV_{MUSE} system exhibited fast kinetics, inducing a significant luciferase signal within 2 hours (**new Supplementary Fig. 7b**). Similar results were verified by qPCR analyses (**new Supplementary Fig. 7c**).

We have added this information in the revised Supplementary Information document. Please see **page 8 and below**:

Supplementary Fig. 7 The kinetic of AAV_{MUSE}-mediated luciferase expression *in vitro*. (a) Schematic for the time schedule and experimental procedure for the muscone-controlled gene expression. HEK-293T cells (6×10^4) transfected with AAV_{MUSE}-encoding plasmids (pWX126, pWX127, pWX158) were cultured for indicated periods (0 to 24 hours) in the presence or absence of 10 μ M muscone. Then cells were collected immediately at the indicated time points (X-axis, 0 to 24 hours) and divided into two groups: one was used to measure luciferase activity (b) and the other one (group B) was collected to extract RNA for qPCR analysis of the luciferase reporter (c). Data are presented as means \pm SD; $n = 3$ biologically independent samples. All plasmids are described in Supplementary Tables S1 and S3.

Secondly, we assessed the kinetics of AAV_{MUSE} *in vivo* by examining the luciferase reporter gene expression. BALB/c mice were intravenously injected with AAV_{MUSE} and exposed to nebulized muscone for 4 hours. By conducting bioluminescence imaging in mice every two hours, we found that 2-hour nebulized muscone exposure was sufficient to trigger luciferase expression, with the maximum luciferase expression at 4 hours after muscone exposure (new Supplementary Fig. 11).

We have added this information in the revised Supplementary Information document. Please see page 12, and below:

Supplementary Fig. 11 Kinetics of AAV_{MUSE}-mediated luciferase expression *in vivo*. (a) Schematic representation of the experimental procedure and the schedule for AAV-2/9 delivery of AAV_{MUSE}-mediated transgene expression in mice. (b) Time-dependent AAV_{MUSE}-mediated transgene expression kinetics in mice. BALB/c mice were injected with the AAV_{MUSE} containing the muscone-responsive vector AAV2/9-pWX126 (3×10^{11} vg), the concatenated $G\alpha_{olf}$ and RTP1S expression vector AAV2/9-pWX127 (1.5×10^{11} vg), and the inducible reporter vector AAV2/9-pWX158 (ITR-P_{CRE}-luciferase-P2A-EGFP-pA-ITR, 1.5×10^{11} vg) via tail vein injection. Two weeks after AAV injection, AAV_{MUSE}-transduced mice were exposed to nebulized muscone using an ultrasonic nebulizer, and bioluminescence imaging was monitored at the indicated time points (-4 hours to 24 hours). (c) Bioluminescence measurements of the luciferase expression based on bioluminescence imaging in b. Data were shown as means \pm SEM ($n = 5$ mice).

We have added the associated methods in the revised “Methods” part. Please see **page 16**, and **below**:

Methods

Luciferase assay

HEK293 cells (6×10^4) were transfected with the AAV_{MUSE} system (pWX126, pWX127, pWX158) and cultured for indicated periods (0 to 24 hours) in the presence or absence of 10 μ M muscone. Then, cells were collected immediately at the indicated time points, and luciferase activity was measured using the Firefly Luciferase Reporter Gene Assay Kit (catalog no. RG005, Shanghai Beyotime Biotechnology Co., Ltd.). The time course of absorbance at 510 nm was measured at 37°C using a Synergy H1 hybrid multimode microplate reader (BioTek Instruments) with Gen5 software (version 2.04).

Quantitative real-time PCR

Tissue samples from the left lateral lobe of the liver were harvested, snap-frozen in liquid nitrogen, and stored at -80°C until use. Total RNA was extracted using an RNAiso Plus kit (catalog no. 9109; Takara Bio), and RNA concentration was measured using NanoDrop OneC (ThermoFisher). A total of 1 μ g of RNA was reverse transcribed into cDNA using a HiScript® III RT SuperMix with the genomic DNA Eraser (catalog no. R323; Vazyme). Real-time PCR was performed on a Real-Time PCR Instrument (Roche, LightCycler 96, Switzerland) with ChamQ Universal SYBR qPCR Master Mix (catalog no. Q711; Vazyme). The following parameters were used for the PCR: 95 °C for 10 min followed by 40 cycles at 95 °C for 30 s, 55 °C for 30 s, and 72 °C for 30 s, and a final extension at 72 °C for 10 min. All samples were normalized to housekeeping gene glyceraldehyde 3-phosphate dehydrogenase (*GAPDH*). Values and the results were expressed as a relative mRNA amount using the standard $2^{-\Delta\Delta C_t}$ method. The sequences of primer pairs used in the study are listed in Supplementary Table 2.

Reviewer #3 (Remarks to the Author):

In the present manuscript Wu et al described the development of a muscone inducible transcription system. The initial data were generated by cell transfection and next the system was evaluated in vivo using AAV as delivery vehicles. The model is quite complex because require the delivery of 3-4 different elements however the data presented indicate that muscone responsive receptor allowed control expression in vitro and in vivo in a dose dependent manner.

Howver there is a question I would like the authors to answer: Can the system be induced by alternative drugs or endogenously produced molecules, did the author tested that?

Reply:

Thanks for this comment.

We have tested multiple endogenously produced molecules that have been reported to trigger cAMP signals: i.e., GLP-1 (Ramos, *J. Gen. Physiol.*, 2008), atrial natriuretic peptide (ANP) (Kübler, *Eur. J.*

Biochem., 1992) and dopamine (Girault, *Arch. Neurol.*, 2004). None of these putative cAMP-agonists activated SEAP production in AAV_{MUSE} transfected cells (**new Supplementary Fig. 3**).

We have added this information in the revised Supplementary Information document. Please see **page 4**, and **below**:

Supplementary Fig. 3 Insensitivity of AAV_{MUSE} to endogenous cAMP-agonists. HEK-293T cells were co-transfected with pWX126, pWX127, and pWX158 and incubated with multiple cAMP-agonists at their physiological concentrations. SEAP levels in the culture supernatants were quantified at 24 hours. Data are presented as means \pm SD; $n = 3$ biologically independent samples. P values were obtained using one-way ANOVA. One-way ANOVA, followed by Dunnett's post hoc test, was used to compare multiple groups. n.s., not significant. All plasmids are described in Supplementary Tables S1 and S3.

Next, we tested the effects of various muscone structural analogs to MOR215-1. The new data showed that two structural analogs (including cyclopentadecanone and cyclopentadecanol) triggered SEAP production, while the non-ligands did not (cross)-trigger its production in MOR215-1/CRE transgenic HEK-293T cells *in vitro* (**new Supplementary Fig. 2**). We have added this information in the revised Supplementary Information document. Please see **page 3** (Refer to our response to major point 3 of reviewer 1).

There are additional data required for a full characterization of the system.

1. Figure 1 a and b should be indicated in the text.

Reply:

We have made this correction in the revised manuscript. Please see **page 3, line 79**.

2. Biodistribution analysis of transgene expression should be analyzed in the different models, AAV9 transduced very efficiently the liver but also other organs and it can even cross the blood brain barrier where Gaolf is expressed. The same for AAV6.

Reply:

Thanks for the suggestion. We have performed new experiments to investigate the biodistribution of transgene expression in AAV_{MUSE}-transduced mice. Two weeks after the AAV injection, AAV_{MUSE}-transduced mice were exposed to nebulized muscone for 4 hours. The liver, brain, heart, spleen, lung, and kidney were dissected and monitored using bioluminescence imaging. We observed a significantly higher luciferase signal in the liver, as opposed to the other organs (**new Supplementary Fig. 9**).

We have added this information in the revised Supplementary Information document. Please see **page 10**, and **below**:

Supplementary Fig. 9 Bioluminescence imaging of mouse tissues. BALB/c mice were transduced with AAV_{MUSE} containing the muscone-responsive vector AAV2/9-pWX126 (3×10^{11} vg), the concatenated $G\alpha_{olf}$ and RTP1S expression vector AAV2/9-pWX127 (1.5×10^{11} vg), and the inducible reporter vector AAV2/9-pWX158 (ITR-P_{CRE}-luciferase-P2A-EGFP-pA-ITR, 1.5×10^{11} vg) via tail vein injection. Two weeks after the AAV injection, AAV_{MUSE}-transduced mice were exposed to nebulized muscone for 4 hours using an ultrasonic nebulizer. Then, the mice were euthanized, and the liver, brain, heart, spleen, lung, and kidney were dissected for bioluminescence imaging. Representative bioluminescence images of mouse tissues from the AAV_{MUSE}-transduced mice were shown.

3. Luciferase, FDG21 and IL-4 should be analyzed in different organs.

Reply:

Thanks for the suggestion. AAV_{MUSE}-mediated luciferase expression was monitored using an *in vivo* imaging system. We observed a significantly higher luciferase signal in the liver than in other organs (**new Supplementary Fig. 9**).

As Δh FGF21 and Δm IL-4 are secreted proteins, we measured their levels in plasma using the corresponding ELISA kit (**Figure 4b** and **Figure 6b**).

4. Luciferase expression decreases with time, is this due to the development of immune reaction?

Reply:

We agree that luciferase expression decreases with time due to the potential immune reaction of AAVs. This similar phenomenon was reported by Martin H. Kang (Kang et al., *Nat. Commun.*, 2020). Indeed, neutralizing antibodies (NAbs) against the AAV capsids posed substantial barriers to effective gene delivery and persistent gene expression, limiting AAVs used for clinical application (Jeune et al., *Hum. Gene Ther. Methods*, 2013).

Several promising strategies were reported to mitigate potential immune reactions and enhance the stability and longevity of gene expression, such as extracellular vesicle-encapsulated AAVs (EV-AAVs) offering higher NAb resistance (Li et al., *Circulation*, 2023); alternative AAV serotypes or engineered rAAV capsid with a lower immune response (Tse et al., *Proc. Natl. Acad. Sci. USA*, 2017).

5. Please indicate how long after muscone exposure DFGF21 is analyzed.

Reply:

Thank you for your suggestion. We have added the information “The Δ hFGF21 levels in plasma were quantified 4 hours after muscone exposure using a human FGF21 ELISA kit every two weeks over 21 weeks.” in the revised manuscript (**page 38**).

6. Could the authors explain why they don't observe expression when using AAV-lung and why they observe expression when using two different serotypes AAV-lung and AAV6. Does AAV6 go to the lung after iv expression. The explanation: “We speculated that the numbers or capsids of the AAV vectors affected gene delivery efficiency” is really very strange.

Reply:

Thanks for your comments. It has been demonstrated previously that the use of a combination of AAV serotypes could facilitate efficient gene delivery for better gene expression (Aubert et al., *Nat. Commun.*, 2020). Initially, adult BALB/c mice were transduced with luciferase reporter using three AAV serotypes (AAV2/6 alone, AAV2/lung alone, and a combination of AAV2/6 and AAV2/lung). Two weeks after the AAV injection, the bioluminescence signal was quantified using an *in vivo* imaging system. Consistent with previous findings, our results showed that the combination of AAV2/6 and AAV2/lung resulted in higher luciferase expression in mouse lungs than AAV2/6 alone or AAV2/lung alone (**new Supplementary Fig. 21**).

Consequently, we combined two AAV serotypes to deliver AAV_{MUSE}: pWX126 packaged into an AAV2/lung capsid and pWX342 packaged into an AAV2/6 capsid. Two weeks after AAV injection,

AAV_{MUSE}-transduced mice were exposed to nebulized muscone for 4 hours every six weeks or exposed to a vehicle (a mixture of castor oil and ddH₂O) (Fig. 5a, b). We found that AAV_{MUSE}-transduced mice exposed to nebulized muscone exhibited significantly higher lung bioluminescence signal intensities (up to 17-fold induction) compared to vehicle-exposed control mice transduced with AAV_{MUSE}. (Fig. 5c, d).

We have corrected it to “We speculated that the number of AAV_{MUSE} component vectors or the capsid of the AAV vectors might have affected gene delivery efficiency” in the revised manuscript. Please see page 9, line 242.

We also included the information in the revised Supplementary Information document. Please see page 27, and below:

Supplementary Fig. 21 Assessing the impact of AAV serotype combinations on luciferase expression in mice. The BALB/c mice were transduced with luciferase reporter using AAV2/6 (2×10^{11} vg), or AAV2/lung (2×10^{11} vg), or dual AAV serotype combination (AAV2/6, 1×10^{11} vg and AAV2/lung, 1×10^{11} vg) by tail vein injection. (a) *In vivo* bioluminescence images of the BALB/c mice transduced with three different AAV vectors packaging luciferase (ITR-PCMV-luciferase-pA-ITR) at 2 weeks after the AAV injection. (b) Bioluminescence measurements of the luciferase expression based on bioluminescence imaging in a. Data in b are presented as means \pm SEM ($n = 3$ mice). P values were obtained from two-tailed unpaired t -tests. **** $P < 0.0001$. Source data are provided as a Source Data file.

7. In the case of asthma the authors should indicate that they prevent the development of sensitivity to ova, they are not treating a chronic disease.

Reply:

Thanks for this comment. We have corrected “AAV_{MUSE-ΔmIL-4} can remotely control ΔmIL-4 to inhibit eosinophil infiltration and pro-inflammatory cytokines and to prevent the development of sensitivity to OVA in allergic asthma mice based on inhalation of muscone.” in the revised manuscript. Please see **page 11**.

Reviewer #4 (Remarks to the Author):

The manuscript reports development and preliminary validation of a very interesting inducible transgene expression system. While the system appears to be quite functional and thus has clear applications in gene therapy applications, there are number of major limitations of the system that, in opinion of this reviewer, significantly limit its translational potential. While the system can be utilised in preclinical studies, the lack of direct translability will limit the impact on the broader field of gene therapy.

Reply:

Thank you for your valuable feedback on our manuscript. We appreciate your interest in our work and acknowledge your concerns. Below, we provide our response to the major limitations you've pointed out:

We completely understand your concerns regarding the limitations of our study. We have explicitly mentioned these limitations in the manuscript and discussed their potential impact in the discussion section. We hope readers will understand that while our research has certain constraints, it still holds the potential to yield valuable outcomes in specific contexts.

You mentioned challenges in the direct translational aspect of our system. We also acknowledge this and have emphasized the current research stage in our discussion. Our primary focus is to provide a useful tool for basic research and preclinical studies, with insights that can benefit the broader field of gene therapy. Although applications at the present stage are limited, we believe this system can be further improved and optimized for enhanced translational potential in future studies.

We have included this information in the revised manuscript. Please see **page 13 and below**:

“Moreover, multiple components of our system are an obstacle for wide application. Possible solutions include designing a smaller version of MUSE suitable for packaging within a single AAV by truncating and mutating the AAV_{MUSE} modules containing RTP1S and MOR215-1, perhaps based on artificial intelligence (AI)-guided protein structure prediction and classification or via high-throughput ligand screening. Another idea for the efficient *in vivo* delivery of MUSE is using engineered virus-like particles (eVLP) (Banskota et al., *Cell*, 2022), which hold great promise as vehicles for targeted therapeutic macromolecules delivery and represent an improvement over AAV or plasmid delivery.”

In addition to the major concerns outlined below, there are also number of minor concerns / errors that need to be addressed.

Major Concerns

While the authors cite in the introductions examples where transgene expression could (did) lead to undesired toxic effects, and use those examples to justify the need to an inducible system, such toxicity has only been observed in a very small subset of gene therapy applications. Thus, while the described system is indeed quite interesting and could have very useful applications in genetic engineering, the clinical need for such system is quite unclear to the reviewer.

Reply:

Thanks for your comments. Our newly developed AAV_{MUSE} system offers a valuable tool that achieves controllable expression of the therapeutic gene. The gene was induced via inhalation of the muscone with a fascinating and pleasant odor used in perfumes, providing needle-free aromatherapy and suitable for specific scenarios where precise control of gene expression is critical and highly beneficial. For instance, AAV2/9-delivered *survival motor neuron (SMN)* overexpression results in sensorimotor toxicity and reverses the initial benefits of spinal muscular atrophy (SMA) therapy (Van Alstyne et al., *Nat. Neurosci.*, 2021). In addition, AAV-delivered *fibroblast growth factor 21 (FGF21)* overexpression results in bone loss owing to constitutive gene expression over physiological concentrations (Wei et al., *Proc. Natl. Acad. Sci. U. S. A.*, 2012). Another example is that excessive erythropoietin (Epo), a hormone widely used to treat anemia, induces potentially fatal polycythemia (Oliveira Junior et al., *J. Bras. Nefrol.*, 2015). To minimize the side effects of these therapeutic proteins, it is essential to equip the AAV with a regulatory system, and the ideal system would allow gene expression to be controlled pharmacologically by the inducers. Our AAV_{MUSE} system may represent one of the ideal control systems, which titrates the therapeutic proteins into the desired range and adjusts their levels as required.

In addition, AAV_{MUSE} could be transduced into target tissues, which can serve as a bio-factory to control various therapeutic protein production (including enzymes, peptide hormones, and even antibodies for treating various diseases) via adjusting muscone dose and exposure time, compared to traditional drugs that are non-adjustable. We considered that this approach could provide robust,

tunable, and on-demand therapeutic proteins for the long-term management of chronic diseases, with the option to terminate patient-selected drugs to prevent drug tolerance or addiction.

Importantly, the administration route of muscone through intermittent applications of nebulized muscone ensures easy clinical implementation and does not affect patients' routines. Future clinical applications of AAV_{MUSE}-based aromatic gene therapy may foresee night-time exposure of patients to muscone vaporized by an ultrasonic aroma diffuser or inhalation of the muscone aroma using modified electronic cigarettes providing portable, lifestyle-compatible and personalized treatment.

In addition, if such system was utilised in a clinical setting, the difficulty in finding the most optimal timing of muscone exposure (and dosing) would create a significant translational problem. This would be a major safety barrier, as 1) too much exposure could lead to transgene expression similar to that observed for a canonical gene augmentation strategies (which, as per authors' opinions, could lead to toxicity); 2) too little expression could make the therapy ineffective / suboptimal.

Reply:

Thanks for your comments. Currently, our designed AAV_{MUSE} is merely a proof-of-concept. The data collected from both cell culture and mouse experiments strongly support that the AAV_{MUSE} system can regulate gene expression based on the control of nebulized muscone concentration or exposure time (**Fig. 2b, c; Fig. 3b, c; new Supplementary Figures 7 and 11**).

For future clinical applications of the AAV_{MUSE} system, more preclinical trials need to be done, including the optimal inducer dosing for desired gene expression levels via adjusting the dose, nebulization frequency, and total muscone nebulization concentration. In addition, we could control our dosing regimen based on control of nebulized muscone concentration or exposure time using non-invasive respiratory (Aerogen® Solo Product), which has been used in the world's first inhalation-based adenovirus vector COVID-19 vaccine (Huang et al., *Lancet Respir. Med.*, 2023).

Nowadays, inhalation therapy has been used for drug delivery and immunization against disease (Lokugamage et al., *Nat. Biomed. Eng.*, 2021). Recent developments have made reliable, portable device(s) for specific groups of individuals.

Finally, (and the authors pointed this as one of the current limitations) the system requires co-delivery of 2 or 3 AAV vectors. This would greatly increase the cost of therapies and would trigger safety concerns. Thus, the translational impact of the technology developed and validated in this study is limited, at least in the foreseeable future.

Reply:

We agree with the reviewer that multiple components in our system hinder its wide application in future translation studies. An ideal solution is to develop a more packable device with a small construct size, ensuring efficient delivery *in vivo*. In the future, artificial intelligence (AI)-guided protein structure prediction and classification will be used to predict the structures of MUSE modules: RTP1S and MOR215-1 and generate the miniature variants with good responsive capability to muscone (Huang et al., *Cell*, 2023). In addition, a smaller version of MUSE suitable for packaging within a single AAV will be engineered by truncating and mutating the MUSE modules via high-throughput ligand screening (Xiang et al., *Nat. Biotechnol.*, 2023 and Yasi et al., *Curr. Opin. Biotechnol.*, 2020). Another idea for the efficient *in vivo* delivery of MUSE is using engineered virus-like particle (eVLP) developed by David R Liu (Banskota et al., *Cell*, 2022), which hold great promise as vehicles for the targeted delivery of therapeutic macromolecules, representing an improvement over AAV or plasmid delivery.

We have included this information about the potential for AAV_{MUSE} system improvement and optimization in the revised “Discussion” part. Please see **page 13** (Please also refer to our response to point 1 of reviewer 2) and **below**:

“Moreover, multiple components of our system are an obstacle for wide application. Possible solutions include designing a smaller version of MUSE suitable for packaging within a single AAV by truncating and mutating the AAV_{MUSE} modules containing RTP1S and MOR215-1, perhaps based on artificial intelligence (AI)-guided protein structure prediction and classification or via high-throughput ligand screening. Another idea for the efficient *in vivo* delivery of MUSE is using engineered virus-like particles (eVLP) (Banskota et al., *Cell*, 2022), which hold great promise as vehicles for targeted therapeutic macromolecules delivery and represent an improvement over AAV or plasmid delivery.”

Minor Comments

1. Line 21: “...control of therapeutic outputs...” – the term “therapeutic outputs” is unclear. Please rewrite the sentences to clarify that it is referring to control of “therapeutic transgene expression”, as the reviewer assumes that is what this term is referring to.

Reply:

We have corrected to “therapeutic transgene expression” in the revised manuscript. Please see **page 1, line 26**.

2. Line 26: Just few sentences later the term “control of transgene expression” is used to what I believe is the same process as in the first sentence. Just like the first term, this expression is also unclear and

needs to be modified to avoid ambiguity.

Reply:

We apologize for this ambiguity. We have corrected to “control of luciferase expression” in the revised manuscript. Please see **page 2, line 32**.

3. Line 25: “AAVMUSE enables remote, muscone dose- and exposure-time-dependent control of transgene expression in livers or lungs of mice for at least 20 weeks, as activated by muscone, a macrocyclic compound of musk.” – no need to state twice that the control is related to “muscone dose” and “activated by muscone” as to this reviewer it is redundant.

Reply:

We have deleted the redundant description “as activated by muscone, a macrocyclic compound of musk” in the revised manuscript. Please see **page 2, lines 31-32, and below**:

“AAV_{MUSE} enables remote, muscone dose- and exposure-time-dependent control of luciferase expression in the livers or lungs of mice for at least 20 weeks.”

4. Line 31: “...can support future gene therapies...” – this is an overstatement. The odorant-molecule-controlled system has not supported any gene therapies, so it cannot support “further” therapies...

Reply:

We appreciate the reviewer's feedback and have revised our statement: “Our odorant-molecule-controlled system can advance gene-based precision therapies for human diseases.” Please see **page 2, lines 36-37**.

5. Line 42: “...adeno-associated viruses (AAVs) are gaining popularity as a platform for gene delivery to treat a variety of human diseases,...”. Viruses are not used to deliver genes (transgenes) to treat diseases. We use “vectors” for that. Please rewrite the sentence to state: “Owing to low immunogenicity and lack of pathogenicity, recombinant vectors based on adeno-associated viruses (rAAVs) are gaining popularity...”

Reply:

Thank you for this helpful suggestion. We have revised the sentence accordingly: “Owing to low immunogenicity and lack of pathogenicity, recombinant vectors based on adeno-associated viruses (AAVs) are gaining popularity...” in the revised manuscript. Please see new **page 2, line 47**.

6. Line 46: the term “recombinant proteins” is used twice in the same sentence. Please rewrite.

Reply:

Sorry for this mistake. We have corrected it to “which can obviate the need to manufacture and frequently inject short-lived recombinant proteins” in the revised manuscript. Please see **page 2, line 50**.

7. Line 66: Please replace “controllable gene therapy” with “controllable expression of gene therapeutic”, to clarify that it is the level of transgene expression that is being controlled.

Reply:

We have corrected it to “controllable expression of therapeutic genes” in the revised manuscript. Please see **page 3, line 71**.

8. Figures should be cited in order of their appearance. While Figure 1 is called in Line 74, the next reference is to Figure 1c (Line 88). Figure 1a and 1b are not referred to.

Reply:

Sorry for this mistake. We have corrected to “**Fig. 1a, b**” as suggested. Please see **page 3, line 79**.

9. Line 151: The authors state that the effects were observed for up to 28 weeks in liver or even 31 weeks (line 195). In the lung the results are reported up to week 20. In the abstract the authors state that “...control of transgene expression in livers or lungs of mice for at least 20 weeks”, while in the discussion (line283) it is stated that “...can induce reporter gene expression after a single AAV administration over 20 weeks”. While none of those statements is actually wrong, it would be probably beneficial if the authors could be more consistent with their conclusion(s).

Reply:

Thanks for this helpful suggestion. We have revised the sentence as follows: “in mouse livers for up to 28 weeks upon muscone-mediated aromatherapy”, “in lungs for up to 20 weeks”, “can induce reporter gene expression for at least 20 weeks after a single AAV administration,” in the revised manuscript. Please see **pages 7, 10, and 12, lines 184, 271, and 323**.

10. Line 209: “We speculated that the numbers or capsids of the AAV vectors affected gene delivery efficiency.” What does the term “numbers” refer to? Why is “capsids” plural, since to the reviewer’s understanding the same capsid was used for all three constructs.

Reply:

Thank you for your valuable feedback. We have revised the sentence accordingly: “We speculated that the number of AAV_{MUSE} component vectors or the capsid of the AAV vectors might have affected gene

delivery efficiency.” Please see **page 9, lines 242-243**.

11. Line 214: “We eventually selected concatenated construct pWX325 (ITR-PSV40-RTP1S::pA-SEAP-PCRE-ITR).” – can the authors provide more information as to what “eventually” led to the selection of pWX325 over pWX322.

Reply:

Thank you for your feedback. To clarify the selection process, we initially optimized the AAV_{MUSE} by simplifying its components to enhance lung transduction efficiency. Even after removing the G_αoif component, the AAV_{MUSE} remained responsive to muscone, as shown in **Supplementary Fig. 1**.

We then explored various combinations of MOR215, RTP1S, and the reporter element, integrating them into two AAV vectors (**Supplementary Fig. 20**). Subsequently, we constructed two types of concatenated vectors for incorporating two expression cassettes: pWX322 (ITR-P_{SV40}-RTP1S-pA::P_{CRE}-SEAP-pA-ITR) and pWX325 (ITR-P_{SV40}-RTP1S::pA-SEAP-P_{CRE}-ITR).

Upon evaluation, the pWX126/pWX325 combination exhibited a higher 21.5-fold induction of SEAP expression and lower background signal, outperforming the pWX126/pWX322 combination, which showed an 8.4-fold induction (**Supplementary Fig. 20a, b**). Consequently, we selected the pWX126/pWX325 combination for further experiments (**Supplementary Fig. 20c and Figure 5**).

We have added the sentence “Compared to pWX126/pWX322, the pWX126/pWX325 combination exhibited a higher 21.5-fold induction of SEAP expression and lower background signal.” in the revised manuscript. Please see **page 9, line 247**.

12. Line 215-217: please make it clear that this refers to a transient transfection of plasmid constructs and now AAV transduction. It may be useful to clarify statement in line 211: “Seeking to facilitate in vivo delivery, we first simplified AAV_{MUSE} into two separate AAV vector plasmids...”

Reply:

Sorry for this ambiguity. We have corrected it to “we first simplified AAV_{MUSE} into two separate AAV vector plasmids” in the revised manuscript. Please see **page 9, lines 244-245**.

13. Lines 219-225: it is unclear to the reviewer why the authors had to use two AAV variants (AAV2/lung and AAV2/6). Both constructs (pWX126 and pWX342). Have to transduce the same cell for this system to work. Why was that not possible using the same AAV variant for both constructs (either AAV2/lung or AAV2/6)?

Reply:

Thanks for this comment. We have performed a new experiment to demonstrate that there was more efficient transduction *in vivo* using two AAV variants (AAV2/lung and AAV2/6) (**new Supplementary Fig. 21**). It has been demonstrated previously that the use of a combination of AAV serotypes could facilitate efficient gene delivery for better gene expression (Aubert et al., *Nat. Commun.*, 2020). Initially, adult BALB/c mice were transduced with luciferase reporter using three AAV serotypes (AAV2/6 alone, AAV2/lung alone, and a combination of AAV2/6 and AAV2/lung). Two weeks after the AAV injection, the bioluminescence signal was quantified using an *in vivo* imaging system. Consistent with previous findings, our results showed that the combination of AAV2/6 and AAV2/lung resulted in higher luciferase expression in mouse lungs, compared to AAV2/6 alone or AAV2/lung alone. Please also refer to our response to point 5 of reviewer#2 and point 6 of reviewer#3.

We would again like to thank all the Reviewers for the positive assessment and the helpful suggestions to improve the quality of this manuscript.

REVIEWER COMMENTS

Reviewer #1 (Remarks to the Author):

The authors responded to my comments appropriately.

Reviewer #2 (Remarks to the Author):

The revised version of the manuscript by Wu et al. represents a much complete an improved version having tackled most of the issues raised by the reviewers.

In particular, they have tackled the two main aspects I was critical to, namely, the first one is that the system contains several, too many components which might affect the usability, reliability and robustness, and the the second aspect is that the system is not orthogonal to the animal cells used, as it hi-jacks partially endogenous central signalling components and pathways which might cause side effects. For the former, the authors have now a discussion on the issue.

With respect to the lack of orthogonality and potential crosstalk with endogenous pathways, in experiments with other receptors inducing main pathways and RNASeq of cells with the MUSE system (induced and uninduced) no evident crosstalk is observed. It is interesting and somehow puzzling that despite the MUSE system triggering cAMP signalling (which is rather a signal induced by and affecting different regulatory networks) no other pathway seems to be activated. The authors should discuss this thoroughly, perhaps supported by literature, on how the "cAMP signal hub" is able to decode the identity of the signal/receptor and activate only its cognate response, being such a general secondary messenger it would be expected that several targets responsive to cAMP would be activated. This is of high relevance for the article and application of the system, and design of future follow up systems based on this principle.

Minor points:

- In Fig 1c. For clarity it would be perhaps useful to separate from GOI the two independent targets (FGF21 and IL-4 towards right and left in correspondence to the mouse models NAFLD and asthma (left and right above).

- In the text added now for the new suppl. Fig 5 (crosstalk), clarify the goal of this experiment: crosstalk meant as activation of the downstream targets of the engineered pathway upon activation of other pathways using general signalling. This is different to the goal of the new experiment fig suppl 15 which is meant to analyse what other pathways might the cAMP generated by the MUSE system be activating/influencing. Also make it clearer in the title and legend of figs suppl 5 and suppl 15. Tune down also the outcomes, there were just 3 pathways tested/assayed, which is not enough to make such a generalized statement.

Reviewer #3 (Remarks to the Author):

The authors have made a commendable effort to address the reviewers' comments. However, there remain several points that are either inadequately addressed or reflect an incorrect interpretation of the published data, which should be addressed before publication.

Reviewer 3, Points 2 and 3:

The authors did not satisfactorily address the biodistribution experiment. The presented luciferase experiment that indicates in vivo luciferase expression solely in the liver is inconclusive. This is because other organs might also be expressing luciferase, but due to the overwhelming expression from the liver, they aren't detected. Both luciferase protein and mRNA levels should be quantified ex vivo, especially in organs where the inducer is detected. Additionally, a similar approach should be considered for the analysis of secretable proteins, since mRNA expression can be assessed in different organs. Given that the AAVs are not organ-specific and the inducible system isn't cell-specific (and considering that muscone after inhalation can reach various organs), one would expect transgene expression in multiple organs. If this isn't the case, what is the rationale behind such phenomena?

Reviewer 3, Point 4:

The authors seem to have misapplied prior work. Specifically, the study by Kang et al., Nat. Commun., 2020 primarily focuses on the therapeutic vector for a severe lung disease, not liver disease. Additionally, while antibodies can prevent initial AAV-transduction, they aren't involved in the reduction of AAV-mediated expression in the liver once it's established.

Reviewer 3, Comment Point 6 and Reviewer 2, Point 5:

The provided rationale for the absence of luciferase expression is not compelling. The authors should consider running an experiment where they administer the three vectors, replacing the vector expressing luciferase in an inducible manner with one expressing luciferase constitutively. This would offer more clarity.

Regarding the experiment with the simplified vector:

The authors seem to have misconstrued the findings from Aubert et al. The referenced study combined different serotypes carrying identical constructs, leading to enhanced outcomes because this strategy allowed targeting of distinct neuron types in various anatomical brain regions. The enhancement was not due to one serotype augmenting the transduction of another.

The excerpt from Aubert et al reads: "Taken together, these data suggest that in order to maximize HSV gene therapy in both SCG and TG, meganuclease delivery may benefit from a combination of different AAV serotypes to optimally target all HSV-infected neurons in both autonomic and sensory ganglia."

Additionally, in Supplementary Fig. 21, the difference between AAV2/lung alone and its combination with AAV6 is twofold. This suggests that AAV2/lung on its own can significantly express luciferase. However, in Supplementary Fig. 19, even with a higher total dose of three vectors from the inducible system, there's a total lack of luciferase expression. These findings are perplexing, and the authors need to investigate if the different components reached the lung cells and, if so, identify why they aren't functioning optimally. Also, the luminescence scale in Supplementary Fig. 19 is conspicuously missing. At the lowest dose, a signal appears in one of the animals; possibly the kinetics in the lung differ.

Answers to Other Reviewers:

The main message from Supplementary Fig. 5 seems ambiguous. Is it trying to convey that IR, TRPM8, or TLR2, when incubated with their respective agonists (insulin 20 ng/mL; menthol, 50 μ M; CU-T12-9, 1 μ M), do not alter the inducibility of the muscone inducible system? If so, this seems counterintuitive to the original query, which was if the muscone-system modifies the response to these signals.

In Supplementary Fig. 11, the choice to begin the kinetic analysis at time -4 is puzzling. What does time 0 represent?

Reviewer #4 (Remarks to the Author):

The reviewer thanks the authors for addressing reviewer's comments and making suggested changes, including additional studies. It required a lot of work, and the reviewer really appreciates it. The outcome is a substantially stronger manuscript, which the reviewer believes will be of interest to the readership of Nature Communications.

Manuscript number: NCOMMS-23-16959A—Point-by-point responses to referees' comments:

Reviewer's Comments:

Reviewer #1 (Remarks to the Author):

The authors responded to my comments appropriately.

Reply:

We are happy that this reviewer is satisfied with our revision work. Thank you.

Reviewer #2 (Remarks to the Author):

The revised version of the manuscript by Wu et al. represents a much complete an improved version having tackled most of the issues raised by the reviewers.

In particular, they have tackled the two main aspects I was critical to, namely, the first one is that the system contains several, too many components which might affect the usability, reliability and robustness, and the the second aspect is that the system is not orthogonal to the animal cells used, as it hi-jacks partially endogenous central signalling components and pathways which might cause side effects. For the former, the authors have now a discussion on the issue.

With respect to the lack of orthogonality and potential crosstalk with endogenous pathways, in experiments with other receptors inducing main pathways and RNASeq of cells with the MUSE system (induced and uninduced) no evident crosstalk is observed. It is interesting and somehow puzzling that despite the MUSE system triggering cAMP signalling (which is rather a signal induced by and affecting different regulatory networks) no other pathway seems to be activated. The authors should discuss this thoroughly, perhaps supported by literature, on how the “cAMP signal hub” is able to decode the identity of the signal/receptor and activate only its cognate response, being such a general secondary messenger it would be expected that several targets responsive to cAMP would be activated. This is of high relevance for the article and application of the system, and design of future follow up systems based on this principle.

Reply:

Thank you for your insightful feedback. We agree with this reviewer's opinion. In response to the reviewer's inquiry, we acknowledge the importance of elucidating why the MUSE system, despite triggering cAMP signaling, does not activate additional signaling pathways. Our speculation is grounded in the transient nature of cAMP signaling activation. We propose that the main reason lies in the transient activation of the cAMP signaling pathway induced by the MUSE system. Consequently, the related targets responsive to cAMP may not sustain activation, making them challenging to detect by ESGA. Supporting this, literature indicates that intracellular cAMP levels are elevated within a few seconds (Wang et al., *Nat Commun.* 2022; Kawata et al., *Proc Natl Acad Sci.* 2022).

Our data revealed no evident crosstalk between the cAMP signaling pathway activated by MUSE and three other signaling pathways (**Supplementary Fig. 5**). The RNASeq analysis indicated that other

potential pathways may remain unaffected by cAMP generated through the MUSE system (**Supplementary Fig. 15**). Several factors contribute to this observation. Firstly, AAV_{MUSE}-transduced mice were exposed to nebulized muscone for 4 hours weekly, resulting in intermittent activation of intracellular cAMP. Secondly, the short half-life of muscone (0.45 h; Li et al., *Tradit. Chin. Drug Res. and Clin. Pharmacol.*, 2000) leads to the transient activation of the cAMP signaling pathway. Lastly, the RNASeq study used frozen liver samples collected 24 hours after muscone induction, contributing to the transient activation of the cAMP signaling pathway.

We have included a concise explanation in the revised manuscript. Please see **page 7**, and **below**:

“...indicating that intermittent and transient stimulation of the MOR215-1 had not detrimental effects on the endogenous target genes of the cAMP signaling pathway due to the transient nature of cAMP signaling activation.”

Minor points:

- In Fig 1c. For clarity it would be perhaps useful to separate from GOI the two independent targets (FGF21 and IL-4 towards right and left in correspondence to the mouse models NAFLD and asthma (left and right above)).

Reply:

Thanks for this helpful suggestion. For clarity, we have modified **Fig. 1c** in the revised manuscript. Please see **page 32**, and **below**:

Fig. 1 Schematic showing the mouse experimental design for AAV_{MUSE}-mediated gene therapy to treat chronic diseases.

- In the text added now for the new suppl. Fig 5 (crosstalk), clarify the goal of this experiment: crosstalk meant as activation of the downstream targets of the engineered pathway upon activation of other pathways using general signaling. This is different to the goal of the new experiment fig suppl 15 which is meant to analyze what other pathways might the cAMP generated by the MUSE system be activating/influencing. Also make it clearer in the title and legend of figs suppl 5 and suppl 15. Tune down also the outcomes, there were just 3 pathways tested/assayed, which is not enough to make such a generalized statement.

Reply:

Thanks for this comment. We apologize for this unclear description. We have corrected the title and detailed the legend of **Supplementary Fig. 5 and 15** in the revised Supplementary Information. Please see **pages 6 and 19**, and **below**:

“Supplementary Fig. 5 Crosstalk analysis between the MUSE-mediated cAMP signaling pathway and three other signaling pathways.”

“Supplementary Fig. 15 Analysis of potential signaling pathways influenced by cAMP induced by the MUSE system through Gene Set Enrichment Analysis (GSEA).”

As directed, we also have tuned down the outcomes. Please see **page 4 and below**:

“...we found no apparent crosstalk between the MUSE-mediated cAMP signaling pathway and three other signaling pathways, including...”

Reviewer #3 (Remarks to the Author):

The authors have made a commendable effort to address the reviewers' comments. However, there remain several points that are either inadequately addressed or reflect an incorrect interpretation of the published data, which should be addressed before publication.

1. Reviewer 3, Points 2 and 3:

The authors did not satisfactorily address the biodistribution experiment. The presented luciferase experiment that indicates in vivo luciferase expression solely in the liver is inconclusive. This is because other organs might also be expressing luciferase, but due to the overwhelming expression from the liver, they aren't detected. Both luciferase protein and mRNA levels should be quantified ex vivo, especially in organs where the inducer is detected. Additionally, a similar approach should be considered for the analysis of secretable proteins, since mRNA expression can be assessed in different organs. Given that the AAVs are not organ-specific and the inducible system isn't cell-specific (and considering that muscone after inhalation can reach various organs), one would expect transgene expression in multiple organs. If this isn't the case, what is the rationale behind such phenomena?

Reply:

Thanks for your comments. The main goal of our study is the design and creation of a MUSE system for controlled gene therapy. It's crucial to clarify that our focus isn't on developing a new AAV vector

for organ specific targeting. In response to this concern, during the initial revision, we dissected various organs and showed, through *in vivo* imaging, that transgene expression was primarily in the liver (**Supplementary Fig. 9**). The reason is that liver cells are susceptible to AAV2/9 infection, allowing co-transfection of the three vectors in a single liver cell.

Regarding the raised point about the biodistribution experiment, we acknowledge that the point does not provide further insights to our overall conclusion. As discussed with the editor, we do not perform the additional experiments to further confirm the biodistribution of AAV2/9-delivered transgene expression through quantification of luciferase and Δ hFGF21 mRNA levels in various organs using qPCR.

Regarding the specific delivery our AAV_{MUSE} system to livers, it would be resolved by using liver-specific promoters driving MOR215 or engineering tissue specific AAV capsids through directed evolution and computer-guided design (Wang et al., *Nat. Rev. Drug Discov.*, 2019).

We have included related discussion in the revised “Discussion” section. Please see **page 13** and **below**:

“Further, for the tissue specific delivery of our AAV_{MUSE} system to livers, it can be resolved by employing liver-specific promoters to drive MOR215 and utilizing engineered tissue-specific AAV capsids through directed evolution and computer-guided design.”

2. Reviewer 3, Point 4:

The authors seem to have misapplied prior work. Specifically, the study by Kang et al., *Nat. Commun.*, 2020 primarily focuses on the therapeutic vector for a severe lung disease, not liver disease. Additionally, while antibodies can prevent initial AAV-transduction, they aren't involved in the reduction of AAV-mediated expression in the liver once it's established.

Reply:

Thanks for this comment. We observed that AAV_{MUSE}-induced luciferase expression decreased with time. This phenomenon was also reported in the previous study where the therapeutic factor IX levels in liver delivered by AAV2 were persisted for only ~10 weeks within the normal level, and subsequently decreased (Manno et al., *Nat. Med.*, 2006).

This phenomenon may be attributed to two factors, which are also challenges in the AAV research field.

- (1) The transduced vectors are gradually lost in mitotic cells after AAV transduction, as rAAVs persist in the form of non-replicating circularized episomes (Wang et al., *Nat. Rev. Drug Discov.*, 2019). It is suspected that the transduced cells are selectively lost or destroyed.

- (2) The rAAV protein capsid, rAAV DNA genome, and the foreign transgene products could interact with the host immune system at multiple layers through adaptive immunity, including B cell- and T cell-mediated responses, or through innate immunity via Toll-like receptor 2 (TLR2) and TLR9. This interaction limits the long-term performance and repeated dosage of AAVs for gene therapies. (Bulcha et al., *Signal Transduct Target Ther.*, 2021; Manno et al., *Nat. Med.*, 2006; Li et al., *Nat. Rev. Genet.*, 2020; Mingozi et al., *Blood*, 2013).

We have added content to the results of the revised manuscript. Please see **page 7** and **below**:

“However, AAV_{MUSE}-induced luciferase expression decreased with time, possibly due to rAAVs as non-replicating circularized episomes, gradually leading to the loss of transduced vectors in mitotic cells after AAV transduction³¹. In addition, the introduction of foreign proteins delivered by rAAVs into a mammalian organism may trigger adaptive or innate immune responses, potentially resulting in a reduction of transgene products”

3. Reviewer 3, Comment Point 6 and Reviewer 2, Point 5:

The provided rationale for the absence of luciferase expression is not compelling. The authors should consider running an experiment where they administer the three vectors, replacing the vector expressing luciferase in an inducible manner with one expressing luciferase constitutively. This would offer more clarity.

Regarding the experiment with the simplified vector:

The authors seem to have misconstrued the findings from Aubert et al. The referenced study combined different serotypes carrying identical constructs, leading to enhanced outcomes because this strategy allowed targeting of distinct neuron types in various anatomical brain regions. The enhancement was not due to one serotype augmenting the transduction of another.

The excerpt from Aubert et al reads: "Taken together, these data suggest that in order to maximize HSV gene therapy in both SCG and TG, meganuclease delivery may benefit from a combination of different AAV serotypes to optimally target all HSV-infected neurons in both autonomic and sensory ganglia."

Reply:

We cannot understand this reviewer's request for us to conduct this experiment. If this reviewer wants us to perform MUSE-mediated luciferase expression in mice using AAV2/6 and AAV2/lung, we have clearly demonstrated the data in **Figure 5a-d**. If the reviewer wants us to observe AAV-delivered constitutive luciferase expression in the mouse lung, we have the data in **Supplementary Fig. 21**.

- 1) Inspired by the previous study reported by Aubert et al., we aimed to enhance AAV infection in the mouse lung by combining two different AAV serotypes to deliver AAV_{MUSE}. Indeed, our results showed that the combination of AAV2/6 and AAV2/lung resulted in significantly higher

bioluminescence signal intensities (up to 17-fold induction) in mouse lungs compared to vehicle-exposed control mice (**Fig. 5c, d**). Additionally, during the first round of revision, we conducted further mouse experiments that demonstrated mice transduced with two different serotype AAVs exhibited significantly higher luciferase signals in the lung compared to mice transduced with only one serotype AAV (**Supplementary Fig. 21**). These data strongly suggest that using two different AAV serotypes can efficiently deliver the gene of interest to the mouse lung.

- 2) Further, we utilized only AAV2/lung to package MUSE system and transduced the BALB/c mice with varying titers of AAV2/lung-AAV_{MUSE} through tail vein injection. However, no strong luciferase signal was detected in the lung after exposure to muscone (**Supplementary Fig. 19**), further suggesting that a single AAV serotype alone cannot effectively deliver MUSE into the lung.

For the reference of the findings from Aubert et al., we have revised the relative sentence as follows (also see **page 10**):

“Inspired by the precious reported study⁴², we rationally selected AAV serotype combinations for delivering constitutive luciferase expression vector (ITR-P_{hCMV}-luciferase-pA-ITR) to the lung...”

After discussion with the editor, we decided not to perform further experiment to address the point raised by this reviewer.

4. Additionally, in Supplementary Fig. 21, the difference between AAV2/lung alone and its combination with AAV6 is twofold. This suggests that AAV2/lung on its own can significantly express luciferase. However, in Supplementary Fig. 19, even with a higher total dose of three vectors from the inducible system, there's a total lack of luciferase expression. These findings are perplexing, and the authors need to investigate if the different components reached the lung cells and, if so, identify why they aren't functioning optimally. Also, the luminescence scale in Supplementary Fig. 19 is conspicuously missing. At the lowest dose, a signal appears in one of the animals; possibly the kinetics in the lung differ.

Reply:

Thanks for this comment. In fact, different amounts of genetic modules were delivered by AAV2/lung in Supplementary Figs. 19 and 21.

In Supplementary Fig. 19, three AAV2/lung vectors were used to package AAV_{MUSE} and transduced into BALB/c mice with varying titers of AAV2/lung-AAV_{MUSE} through tail vein injection. However, no significant luciferase signal was detected in the lung after exposure to muscone. This is mainly due to the transduction efficiency, as it is challenging to simultaneously transduce all three AAV vectors

into the same single cell. Consequently, no significant luciferase signal was detected in the lung after exposure to muscone.

In Supplementary Fig. 21, a single AAV2/lung vector was used to package a constitutive luciferase reporter (ITR-P_{hCMV}-luciferase-pA-ITR) and transduced into BALB/c mice. The luciferase signal was detected in the lung, consistent with the report by Körbelin that AAV2/lung mediated strong and specific gene expression in the lung after intravenous administration (Körbelin et al., *Mol. Ther.*, 2016).

We have included the maximum and minimum values to the luminescence scales in the revised **Supplementary Fig. 19**.

5. Answers to Other Reviewers:

The main message from Supplementary Fig. 5 seems ambiguous. Is it trying to convey that IR, TRPM8, or TLR2, when incubated with their respective agonists (insulin 20 ng/mL; menthol, 50 μ M; CU-T12-9, 1 μ M), do not alter the inducibility of the muscone inducible system? If so, this seems counterintuitive to the original query, which was if the muscone-system modifies the response to these signals.

Reply:

Thanks for this comment. To address your concern, we have re-conducted cell culture experiment to study the crosstalk analysis between the MUSE-mediated cAMP signaling pathway and three other signaling pathways. Our results further showed that we found no apparent crosstalk between the MUSE-mediated cAMP signaling pathway and three other signaling pathways, including mitogen-activated protein kinase (MAPK)-mediated protein kinase (IRS-1-Ras-MAPK) pathway through the activation of insulin receptor (IR), intracellular calcium-dependent nuclear factor of activated T cells (NFAT) pathway through the activation of transient receptor potential (TRP) melastatin 8 (TRPM8), and proinflammatory nuclear factor NF- κ B activation in response to toll-like receptor 2 (TLR2) signaling (**new Supplementary Fig. 5**).

Supplementary Fig. 5 Crosstalk analysis between the MUSE-mediated cAMP signaling pathway and three other signaling pathways. (a) Schematic representations of the MUSE-mediated cAMP signaling pathway and the insulin receptor (IR)-mediated MAPK signaling pathways. (b, c) Crosstalk analysis between the MUSE mediated-cAMP signaling pathway and the IR-mediated MAPK signaling pathway. HEK-293T cells were co-transfected with plasmids expressing luciferase under the control of the MUSE-mediated cAMP signaling pathway and plasmids encoding secreted alkaline phosphatase (SEAP) expression under the control of the IR-mediated MAPK signaling pathway. Transfected cells were cultured in the presence or absence of insulin (20 ng/mL) with or without muscone (10 μM). (d) Schematic representations of the MUSE-mediated cAMP signaling pathway and the transient receptor potential (TRP) melastatin 8 (TRPM8)-mediated NFAT signaling pathway. (e, f) Crosstalk analysis between the MUSE-mediated cAMP signaling pathway and the TRPM8-mediated NFAT signaling pathway. HEK-293T cells were co-transfected with plasmids expressing luciferase under the control of the MUSE-mediated cAMP signaling pathway and plasmids encoding SEAP expression under the control of the TRPM8-mediated NFAT signaling pathway. Transfected cells were cultured in the presence or absence of menthol (50 μM) with or without muscone (10 μM). (g) Schematic representations of the MUSE-mediated cAMP signaling pathway and the toll-like receptor 2 (TLR2)-mediated NF-κB signaling pathway. (h, i) Crosstalk analysis between the MUSE-mediated cAMP signaling pathway and the TLR2-mediated NF-κB signaling pathway. HEK-293T cells were co-transfected with plasmids expressing luciferase under the control of the MUSE-mediated cAMP signaling pathway and plasmids encoding SEAP

expression under the control of the plasmids encoding TLR2-mediated NF- κ B signaling pathway. Transfected cells were cultured in the presence or absence of CU-T12-9 agonists (50 μ M) with or without muscone (10 μ M). SEAP and luciferase expression were profiled 24 hours after induction. Data are presented as means \pm SD; $n = 3$ biologically independent samples. P values were obtained from two-tailed unpaired t-tests. n.s., not significant. All plasmids are described in Supplementary Tables S1 and S3.

6. In Supplementary Fig. 11, the choice to begin the kinetic analysis at time -4 is puzzling. What does time 0 represent?

Reply:

Thanks for pointing out the unclear description of time representation. The time range from -4 to 0 h represents the duration of nebulized muscone, with 0 h indicating the cessation of nebulization.

We have added this information in the schematic representation and the legend of **Supplementary Fig. 11**. Please see **page 13** in the revised Supplementary Information.

“The time range from -4 to 0 h represents the duration of nebulized muscone, with 0 h indicating the cessation of nebulization.”

Reviewer #4 (Remarks to the Author):

The reviewer thanks the authors for addressing reviewer's comments and making suggested changes, including additional studies. It required a lot of work, and the reviewer really appreciates it. The outcome is a substantially stronger manuscript, which the reviewer believes will be of interest to the readership of *Nature Communications*.

Reply:

We are happy that this reviewer is satisfied with our revision work. Thanks.

Reviewers' Comments:

Reviewer #3:

Remarks to the Author:

I respect the editorial decision not to request additional experiments. However, I don't think it is correct to state that you were inspired by Aubert et al. Aubert et al. demonstrated increased expression with the combination of two serotypes because each serotype targets different types of cells. Therefore, the signal they observed is the sum of the expression from cells transduced by one vector and the cells transduced with the other vector. In contrast, here, the two vectors need to transduce the same cells to achieve expression. Thus, if the reason for having more expression is because different types of cells are targeted, this is contradictory to the requirement of the inducible system described by the authors that requires that the different components are expressed in the same cell. But maybe I am not properly understanding how the system works, and I apologize.

The possibility that one vector enhances the transduction efficiency of the other vector cannot be discarded, and it will be the first time this synergistic effect has been observed using AAV. It will be very interesting to understand the mechanism, but of course, it is out of the scope of this publication.

Said that, I am not sure that Aubert's work can be used as the reference inspiring the combination of the two vectors for the reasons previously mentioned.

Manuscript number: NCOMMS-23-16959B—Point-by-point responses to referees' comments:

REVIEWERS' COMMENTS

Reviewer #3 (Remarks to the Author):

I respect the editorial decision not to request additional experiments. However, I don't think it is correct to state that you were inspired by Aubert et al. Aubert et al. demonstrated increased expression with the combination of two serotypes because each serotype targets different types of cells. Therefore, the signal they observed is the sum of the expression from cells transduced by one vector and the cells transduced with the other vector. In contrast, here, the two vectors need to transduce the same cells to achieve expression. Thus, if the reason for having more expression is because different types of cells are targeted, this is contradictory to the requirement of the inducible system described by the authors that requires that the different components are expressed in the same cell. But maybe I am not properly understanding how the system works, and I apologize.

Said that, I am not sure that Aubert's work can be used as the reference inspiring the combination of the two vectors for the reasons previously mentioned.

Reply:

Thanks for your comments. After careful consideration and to avoid unnecessary misunderstandings, we have deleted the reference of the findings from Aubert's study and have revised the relative sentence as follows in the main text (also see **page 10**):

“Moreover, we selected AAV serotype combinations for delivering constitutive luciferase (ITR-P_{hCMV}-luciferase-pA-ITR) to the lung via the tail vein and demonstrated that the combination of AAV2/lung and AAV2/6 resulted in higher bioluminescence signal intensities in lungs compared to either AAV2/6 alone or AAV2/lung alone.”

The possibility that one vector enhances the transduction efficiency of the other vector cannot be discarded, and it will be the first time this synergistic effect has been observed using AAV. It will be very interesting to understand the mechanism, but of course, it is out of the scope of this publication.

Reply:

Thanks for your comments. Indeed, our results showed that mice transduced with two different serotype AAVs exhibited significantly higher luciferase signals in the lung compared to mice transduced with only one serotype AAV (**Supplementary Fig. 21**). These data suggest that using two different AAV serotypes can efficiently deliver the gene of interest to the mouse lung.

We think that the potential reason is that different AAV serotypes have varied receptor bindings (see **Table 1**), which might increase co-transfection efficiency of the different vectors into a single cell (Pupo A et al, *Mol. Ther.*, 2022; Mingozi F et al., *Nat. Rev. Genet.*, 2011). The AAV2/lung used in this study is derived from AAV2 variant displaying the ESGHGYF peptide (AAV2-ESGHGYF), which mediated strong and specific gene expression (> 200-fold higher) in the lung though a specific receptor compared to wild-type AAV2 vectors (Körbelin et al., *Mol. Ther.*, 2016).

Table 1 Primary receptor and co-receptors of the different AAV serotypes

AAV serotype	Primary receptor	Co-receptor
AAV6	Heparan sulfate proteoglycan N-linked sialic acid	EGFR, AAVR
AAV2	Heparan sulfate proteoglycan	Integrin, FGFR, HGFR, LamR, AAVR
AAV9	Terminal N-linked Galactose of SIA	LamR, AAVR

However, the underlying mechanism behind the increased expression resulting from the combination of two serotypes, which indicates a promising avenue, will be further investigated in the future studies, extending beyond the scope of this work.

With the advancement of AAV capsid engineering technology, including rational design, directed evolution and computer-guided design of ancestral capsid libraries (Wang D et al, *Nat. Rev. Drug Discov.*, 2019), an increasing number of serotypes specifically targeting the lung can be developed. These will drive the optimal AAV combinations for delivery AAV_{MUSE} system to enhance transduction of MUSE system and facilitate its application in disease treatment.